# The push-to-open mechanism of the tethered mechanosensitive ion channel NompC

Yang Wang[1,2†‡], Yifeng Guo[3†‡], Guanluan Li[3,4], Chunhong Liu[1], Lei Wang[1], Aihua Zhang[1,2], Zhiqiang Yan[3,4]*, Chen Song[1,2]*

[1]Center for Quantitative Biology, Academy for Advanced Interdisciplinary Studies, Peking University, Beijing, China; [2]Peking-Tsinghua Center for Life Sciences, Academy for Advanced Interdisciplinary Studies, Peking University, Beijing, China; [3]State Key Laboratory of Medical Neurobiology and MOE Frontiers Center for Brain Science, Institute of Brain Science, School of Life Sciences, Fudan University, Shanghai, China; [4]Institute of Molecular Physiology, Shenzhen Bay Laboratory, Shenzhen, China

*For correspondence:
zqyan@szbl.ac.cn (ZY);
c.song@pku.edu.cn (CS)

[†]These authors contributed equally to this work
[‡]Affiliations 1, 2 and 3 contributed equally to this work

**Abstract** NompC is a mechanosensitive ion channel responsible for the sensation of touch and balance in *Drosophila melanogaster*. Based on a resolved cryo-EM structure, we performed all-atom molecular dynamics simulations and electrophysiological experiments to study the atomistic details of NompC gating. Our results showed that NompC could be opened by compression of the intracellular ankyrin repeat domain but not by a stretch, and a number of hydrogen bonds along the force convey pathway are important for the mechanosensitivity. Under intracellular compression, the bundled ankyrin repeat region acts like a spring with a spring constant of ~13 pN nm$^{-1}$ by transferring forces at a rate of ~1.8 nm ps$^{-1}$. The linker helix region acts as a bridge between the ankyrin repeats and the transient receptor potential (TRP) domain, which passes on the pushing force to the TRP domain to undergo a clockwise rotation, resulting in the opening of the channel. This could be the universal gating mechanism of similar tethered mechanosensitive TRP channels, which enable cells to feel compression and shrinkage.

## Introduction

Many types of sensations initiate from the gating of transient receptor potential (TRP) ion channels, which regulate the intracellular cation concentration that triggers downstream signaling pathways (*Montell et al., 2002*; *Mutai and Heller, 2003*; *Pedersen et al., 2005*; *Basbaum et al., 2009*; *Cheng et al., 2010a*; *Fowler and Montell, 2013*). NompC is one of the earliest identified mechanosensitive ion channels belonging to the TRP family, which plays crucial roles in the sensation of light touch, hearing, balance, and locomotion of *Drosophila melanogaster* (*Walker et al., 2000*; *Göpfert et al., 2006*; *Yan et al., 2013*; *Zhang et al., 2013*; *Zanini et al., 2018*). NompC is structurally unique as it has the largest number of ankyrin repeats (ARs) among the known TRP channels (*Montell, 2005*), 29 in total. The AR region is associated with microtubules, and it has been proposed to act as a gating spring to regulate the channel gating according to the so-called 'tethered gating model' (*Albert et al., 2007*; *Cheng et al., 2010b*; *Zhang et al., 2015*). Although NompC orthologs have not been found in mammals (*Venkatachalam and Montell, 2007*; *Knecht et al., 2015*), it was shown to function in mechanosensation of *Caenorhabditis elegans* and *Danio rerio* as well (*Kang et al., 2010*; *Sidi et al., 2003*). It therefore serves as a useful model for studying the molecular mechanism of the tethered mechano-gating. The cryo-EM structure of NompC has been

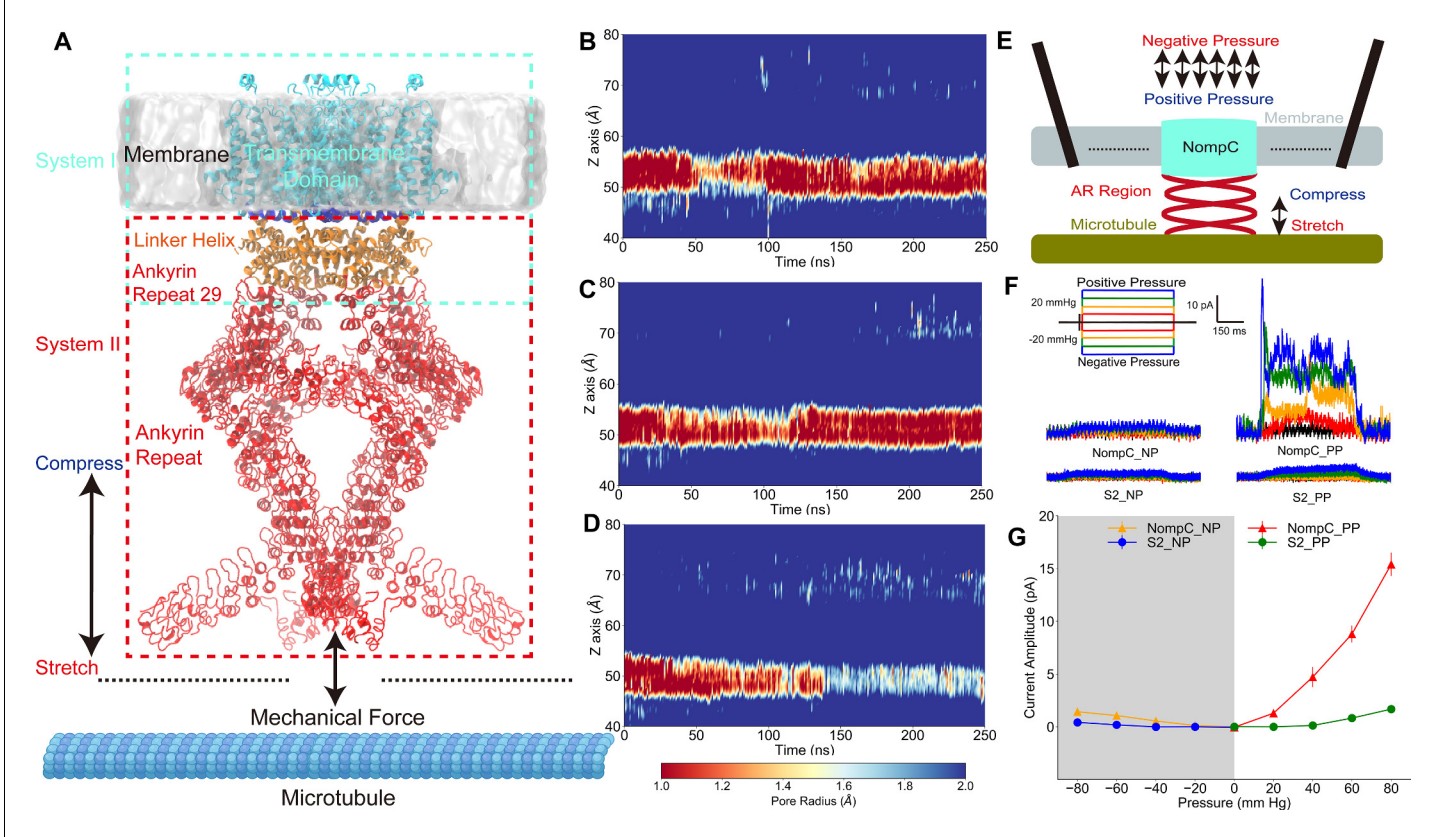

**Figure 1.** The tethered NompC channel was opened by compression of the intracellular ankyrin repeat domain. (A) The simulation systems. The NompC was divided into two subsystems, denoted by the cyan and red rectangular boxes, for the molecular dynamics (MD) simulations. (B−D) The transmembrane (TM) pore size evolution for the force-free (B), pulling/stretch (C), and pushing/compression (D) simulations, calculated from the MD trajectories FI0, SI0, and CI0 (*Supplementary file 1a*), respectively. (E) A schematic figure of the cell-attached patch-clamp electrophysiological experiment for NompC. (F) Representative traces of the electrophysiological measurements for the S2 blank cell and NompC-expressed cell, showing that there are significantly larger signals under positive pressure (PP) in the presence of NompC. (G) The mean and standard deviation (SD) of the mechano-gated currents in the S2 blank and NompC expressing cells under positive (PP) and negative pressure (NP) in the cell-attached patch-clamp experiments (S2_NP: n = 6; S2_PP: n = 6; NompC_NP: n = 5; NompC_PP: n = 7). All of the error bars denote ± SD.

The online version of this article includes the following figure supplement(s) for figure 1:

**Figure supplement 1.** The sequence of NompC used for the molecular dynamics (MD) simulations (highlighted).

**Figure supplement 2.** The transmembrane (TM) pore size evolution of multiple replicate simulations with a slower pulling/pushing speed, for the force-free, pulling/stretch, and pushing/compress simulations, calculated from the molecular dynamics (MD) trajectories FI1-3, SI1-3, and CI1-3 (*Supplementary file 1a*), respectively.

**Figure supplement 3.** This figure is similar to *Figure 1B–D* except that the pore radius was calculated with the structure of the transmembrane (TM) backbone (side chain removed) in the molecular dynamics (MD) trajectories FI0, SI0, and CI0 (*Supplementary file 1a*),respectively.

**Figure supplement 4.** This figure is similar to *Figure 1—figure supplement 2* except that the pore radius was calculated with the structure of the transmembrane (TM) backbone only (side chain removed) in the molecular dynamics (MD) trajectories FI1-3, SI1-3, and CI1-3, respectively.

**Figure supplement 5.** The number of water molecules around the gate region of NompC in the molecular dynamics (MD) simulations.

**Figure supplement 6.** Sodium ions spontaneously permeated through the partially opened gate in the 'pushing' molecular dynamics (MD) simulations in the absence of a transmembrane potential in the trajectories CI1 and CI2, respectively.

**Figure supplement 7.** Ion density maps from the ion permeation trajectories.

**Figure supplement 8.** The sodium ion and potassium ion permeation count through the partially opened structure of NompC in the permeation molecular dynamics (MD) trajectories PI1-3 and PII1-3, respectively.

**Figure supplement 9.** Experiment results of the inside-out (IO) and outside-out (OO) patch clamp.

**Figure supplement 10.** The mechanosensitive current can be blocked by GdCl₃.

**Figure supplement 11.** Distances between the centers of AR29 and the transmembrane (TM) domain of NompC in the 250 ns molecular dynamics/ steered molecular dynamics (MD/SMD) simulations of system I, for the free, pulling, and pushing simulations, calculated from the MD trajectories FI0, SI0, and CI0, respectively.

**Figure supplement 12.** Distances between the centers of AR29 and the transmembrane (TM) domain of NompC in the 500 ns molecular dynamics/ steered molecular dynamics (MD/SMD) simulations of system I, for the free, pulling, and pushing simulations, calculated from the MD trajectories FI1-3, SI1-3, and CI1-3, respectively.

*Figure 1 continued on next page*

*Figure 1 continued*

**Figure supplement 13.** The overlaid initial and 200 ns conformations of the simulation system I in the free, pulling/stretch, and pushing/compress simulations, from the molecular dynamics (MD) trajectories FI0, SI0, and CI0, respectively.

resolved (*Jin et al., 2017*), showing that four AR chains form an ~15-nm-long supercoiled helix and connect to the transmembrane (TM) pore domain via a linker helix (LH) region (*Figure 1A*).

The new structure confirmed that the AR helices probably act as a spring to conduct forces to the TM pore when the neuron cells deform. However, what kind of forces (or what type of cell deformation) can open the NompC channel, and how the force is transduced from ARs to the TM region to finally open the pore, are still elusive. In previous studies, it has been suggested that pulling the AR spring may open the channel (*Zhang et al., 2015*; *Gaudet, 2008*). In contrast, there are other models indicating that a pushing force may be required to open the channel (*Howard and Bechstedt, 2004*; *Argudo et al., 2019*). Therefore, the detailed gating mechanism of this unique tethered ion channel requires clarification. Additionally, the membrane surface tension-induced ion channel gating provides a mechanism by which cells can respond to volume expansion (*Martinac et al., 1990*; *Sukharev, 2002*; *Nomura et al., 2012*; *Zhang et al., 2018*; *Martinac et al., 2018*). However, there is no obvious mechano-gating mechanism that can respond to cell compression or volume shrinkage. In this study, we combined molecular dynamics (MD) simulations and electrophysiological experiments to study the detailed gating mechanism of NompC. We provide a plausible push-to-open mechanism for the tethered ion channels, which may be used by cells to sense and respond to compression and shrinkage.

## Results

### TM pore opens under an intracellular pushing force

To study the atomistic details of how mechanical stimuli can lead to the gating of the tethered NompC channel, we used a divide-and-conquer protocol. We performed all-atom MD simulations on the transmembrane and linker helix (TM + LH) domains, and the linker helix and ankyrin repeat (LH + AR) domains of the cryo-EM NompC structure, respectively (*Figure 1A*, *Figure 1—figure supplement 1*). We considered two forms of the most essential forces on the AR helices: pulling and pushing. For the TM + LH system, we applied forces that are normal to the membrane surface on the AR29, which directly connects to the LH region, and we monitored how the TM domain responds by calculating the radius of the TM pore. We observed that the channel remains closed (with a very narrow constriction site, radius <1.0 Å, around the residue I1554) throughout the simulations without any external forces (*Figure 1B* and *Figure 1—figure supplement 2A*, *Video 1*), indicating that the closed-state cryo-EM structure was stable in our 'force-free' MD simulations. When the direction of the pulling force was away from the membrane surface, the TM channel also remained closed in our MD simulations (*Figure 1C* and *Figure 1—figure supplement 2B*, *Video 1*). In fact, the narrow region

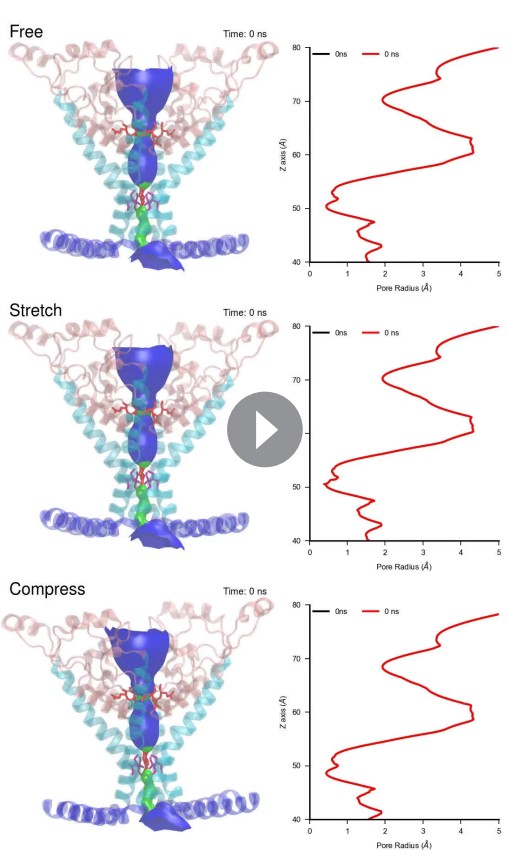

**Video 1.** The transmembrane (TM) pore size evolution during the 250 ns simulation trajectories (FI0, SI0, and CI0) as shown in *Figure 1B–D*. This video shows the TM pore size evolution during the 250 ns 'free', 'pulling', and 'pushing' simulations.
https://elifesciences.org/articles/58388#video1

with a radius of less than 1 Å expanded in the latter part of the trajectory compared to the 'force-free' simulations, indicating that the channel was actually more closed than the free NompC in our 'pulling' simulations. In contrast, when applying a proper pushing force (toward the membrane) on the AR29, we observed that the narrowest constriction site of the channel was significantly dilated in the latter part of the 'pushing' simulation (*Figure 1D* and *Figure 1—figure supplement 2C*, *Video 1*). The above-mentioned analysis was carried out based on the all-atom structure of NompC in the MD trajectories. We also conducted pore radius analysis based only on the backbone structures, and we observed the same trend as mentioned above (*Figure 1—figure supplement 3* and *Figure 1—figure supplement 4*), which confirmed that the pore dilation was due to the global conformational change of the backbone rather than merely a side-chain movement.

The number of water molecules at the gate region should be increased when the pore is dilated, which is often viewed as an additional indicator for the channel opening. Our analysis showed that the number of water molecules was indeed evidently increased in the 'pushing' simulations, as compared to the 'free' and 'pulling' simulations (*Figure 1—figure supplement 5*). In the absence of TM potential, we observed two spontaneous ion permeation events only when the pore was dilated under a pushing force (*Figure 1—figure supplement 6*). When applying a membrane potential of ±300 mV, we observed continuous ion permeation through the dilated pore caused by the pushing force in our MD simulations (*Figure 1—figure supplement 7* and *8*, *Video 2*). Therefore, our simulation results indicated that the NompC channel may be opened by a pushing force from the intracellular side but not by a pulling force.

To validate these findings, we did cell-attached patch-clamp experiments (*Figure 1E*), in which positive or negative pressure with a 20 mm Hg increment was applied. Since the AR region is associated with microtubules (*Liang et al., 2013*), it is conceivable that positive pressure will result in a slight compression of the AR region and thus a pushing force on the TM domain, whereas a negative pressure will generate a slight stretch of the AR region and a pulling force on the TM domain in the cell-attached patch-clamp experiments. As shown in *Figure 1F and G*, the reference *Drosophila* S2 cells without NompC expressed showed no response to the positive and negative pressure stimuli, while we detected a clear electrical signal through the NompC-expressed S2 cells under positive pressure, whereas the signal under negative pressure was nearly negligible. Similarly, we can detect a clear signal through the NompC-expressed S2 cells from the outside-out patch clamp under negative pressure which corresponds to the cell-attached patch clamp under positive pressure. On the other hand, the signal under inside-out patch clamp with negative pressure was nearly negligible (*Figure 1—figure supplement 9*). The electrical signals from cell-attached patch clamp under positive pressure and outside-out patch clamp under negative pressure were nearly completely abolished after adding GdCl₃ (a blocker for NompC) to the bath (*Figure 1—figure supplement 10*), confirming that the detected signal was indeed due to the ion permeation through NompC. Our results are consistent with a previous study showing that NompC can be activated by mechanical forces, and the AR regions are crucial for the mechano-gating of NompC (*Zhang et al., 2015*). Therefore, our experimental results indicated that it is the compression of the AR region and the resulted pushing force that opens the channel, which is consistent with MD simulations.

## Conformational changes of the TM domain associated with gating

We investigated how a pushing force from the AR region can open the TM pore by analyzing the TM + LH simulations. The free, pulling, and pushing trajectories were concatenated, and principal component analysis (PCA) was performed to visualize the collective motion of the NompC pore domain. As shown in *Figure 2A*, the second PCA eigenvector can distinguish the conformations of the free, pushing, and pulling simulations very well, with the larger values corresponding to the more dilated states. We extracted the two extreme conformations along the second PCA eigenvector, and we overlaid them to visualize

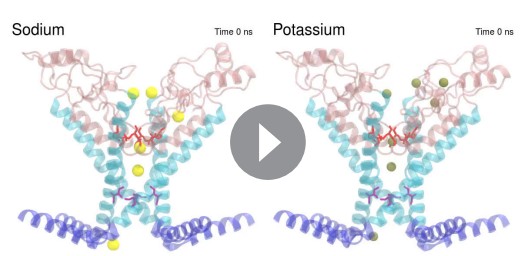

**Video 2.** Ion permeation through the partially opened NompC channel under transmembrane potential.
https://elifesciences.org/articles/58388#video2

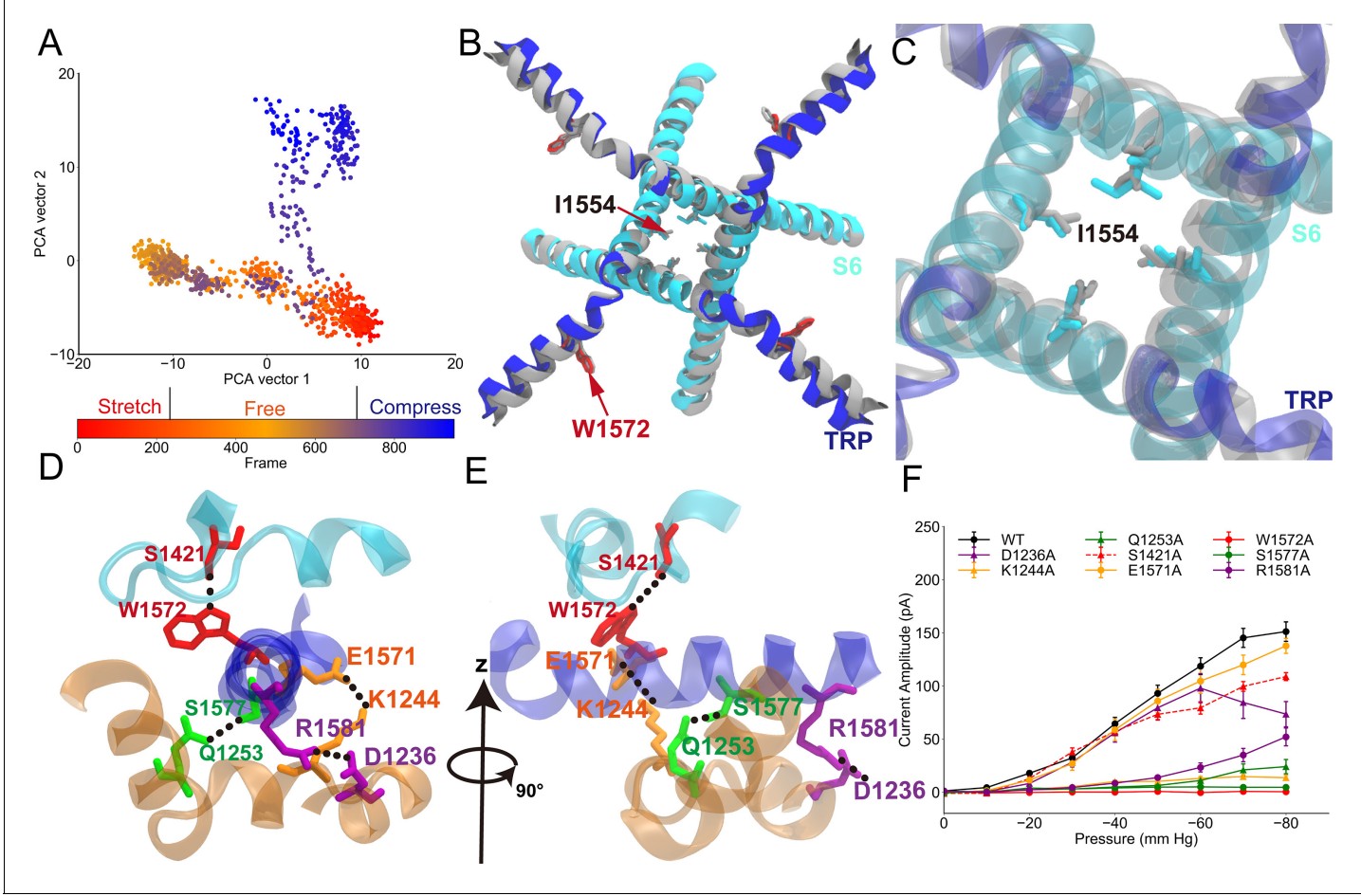

**Figure 2.** Conformational changes of the transient receptor potential (TRP) and transmembrane (TM) domains during gating. (**A**) Principal component analysis (PCA) of the molecular dynamics (MD) simulation trajectories (FI0, SI0, and CI0 in *Supplementary file 1a*). The projections on the second eigenvector can distinguish the conformations under pulling (red) or pushing (blue). (**B**) The overlaid extreme structures along the second eigenvector of the PCA. The most closed conformation (silver) and open conformation (cyan) showed the global changes of the TRP domain during gating: a clockwise rotation. (**C**) The orientation and position of the gate residue, I1554, in the most closed (silver) and open (cyan) conformations in the simulations. (**D, E**) The residues forming four stable hydrogen bonds between the TRP and LH domains. (**F**) The mean and standard deviation (SD) of the mechano-gated current of the wild-type NompC, as well as the mutants W1572A, S1421A, Q1253A, S1577A, K1244A, E1571A, D1236A, and R1581A, under negative pressure in the outside-out patch-clamp experiments (wild type: n = 13; W1572A: n = 7; S1421A: n = 6; Q1253A: n = 6; S1577A: n = 5; K1244A: n = 6; E1571A: n = 6; D1236A: n = 9; R1581A: n = 6). All of the error bars denote ± SD. Hydrogen bonds are indicated by dashed lines (**D, E**).

The online version of this article includes the following figure supplement(s) for figure 2:

**Figure supplement 1.** The transmembrane (TM) pore size evolution and rotation angle evolution of the transient receptor potential (TRP) domain from principal component analysis (PCA).

**Figure supplement 2.** The conformational change of the transient receptor potential (TRP) domain in the steered molecular dynamics (SMD) simulations.

**Figure supplement 3.** The overlaid closed-state and open-state structures of TRPV1, obtained in lipid nanodisc.

**Figure supplement 4.** The schematic figure of pUAST-NOMPC_EGFP (del-miniwhite) used in the experiment.

**Figure supplement 5.** The mutants of residues listed in *Figure 2D–F* showed normal membrane targeting.

**Figure supplement 6.** Formation of the alternative hydrogen bonds in the mutant, as identified in the molecular dynamics (MD) simulations (from chain A in the trajectories D1236A and E1571A in *Supplementary file 1a*).

the most significant conformational changes of the TM domain under the three mechanical stimuli (*Figure 2B*). We observed an evident clockwise rotation (looking from the intracellular side, *Figure 2B*, *Figure 2—figure supplements 1–2*, and *Video 3*) and an obvious upward tilt (looking from lateral side of membrane, *Figure 2—figure supplements 1–2*, and *Video 4*) of the TRP domain when a pushing force was applied to the AR29. This clockwise rotation and upward tilt of the TRP

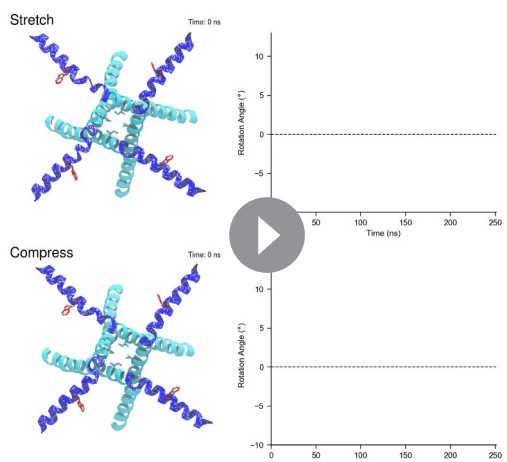

**Video 3.** The rotation of the transient receptor potential (TRP) domain in the steered molecular dynamics (SMD) simulations (SI0 and CI0 in *Supplementary file 1a*).
https://elifesciences.org/articles/58388#video3

domain may be associated with the opening of the TRP channels (*Liao et al., 2013*; *Cao et al., 2013*; *Gao et al., 2016*; *Zheng and Qin, 2015*). The overlaid structures in *Figure 2B* show that the clockwise rotation of the TRP domain induced the S6 helices (which are directly linked to the TRP domain) to rotate clockwise as well, albeit to a lesser extent. The gating constriction site is located at I1554 of the S6 helix (*Jin et al., 2017*), and in our simulations, they were pulled away from the channel axis when the S6 helices rotate clockwise together with the TRP domain, leading to the dilation of the pore (*Figure 2B and C*). Thus, consistent with previous structural studies of TRPV1 (*Figure 2—figure supplement 3*; *Liao et al., 2013*; *Cao et al., 2013*; *Gao et al., 2016*), our simulations showed that the clockwise rotation of the TRP domain (as well as the S6 helices) may lead to the opening of the NompC pore. It is the pushing force (compression of the intracellular domain) that leads to this collective gating motion.

## Key residues around the TRP domain for gating

We analyzed the hydrogen bonding network around the TRP domain, and we attempted to locate the key residues ensuring the clockwise rotation of the TRP domain in response to the pushing force from AR. We identified four stable hydrogen bonds throughout the MD simulations (*Figure 2D and E* and *Supplementary file 1d*). Three of the four hydrogen bonds can also be directly identified in the cryo-EM structure, except for the one between Q1253 and S1577, which was more stable only in the presence of a pushing force in the MD simulations (*Supplementary file 1d*). These stable hydrogen bonds indicate a conservative interaction network as well as a stable local configuration during the gating process. We then did mutations on the residues forming these hydrogen bonds and performed electrophysiological experiments to determine if any of them play crucial roles in the gating of NompC. *Figure 2F* shows that mutations of most of the eight residues led to some loss-of-function. In the meantime, the mutants showed normal membrane targeting (*Figure 2—figure supplement 5*). This indicated that most of the mutations changed the mechanosensitivity of NompC. In particular, the W1572A mutation completely abolished the gating behavior of the channel, consistent with the work of *Jin et al., 2017*. Interestingly, we found that W1572 may be the rotation pivot of the TRP domain in our MD trajectory, which forms a stable hydrogen bond with the backbone of S1421 on the S4-S5 linker. This highlighted the importance of the interactions between the TRP domain and the S4-S5 linker in the gating process. Notably, this hydrogen bond does not involve the side chain of S1421, so the mutations at S1421 would not be expected to alter the above hydrogen bond and would not lead to significant loss-of-function of NompC. This was confirmed for S1421A, as

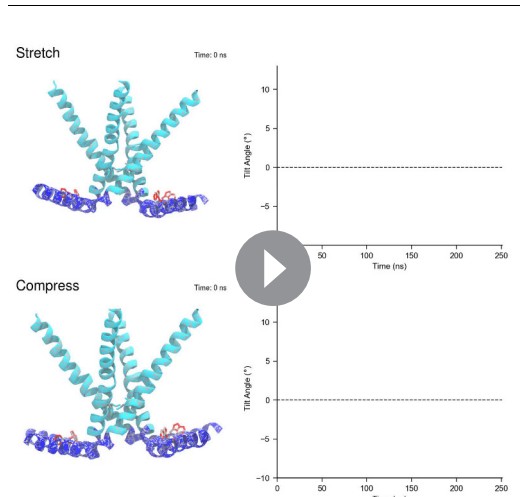

**Video 4.** The tilt of the transient receptor potential (TRP) domain in the steered molecular dynamics (SMD) simulations (SI0 and CI0 in Supplement 1a).
https://elifesciences.org/articles/58388#video4

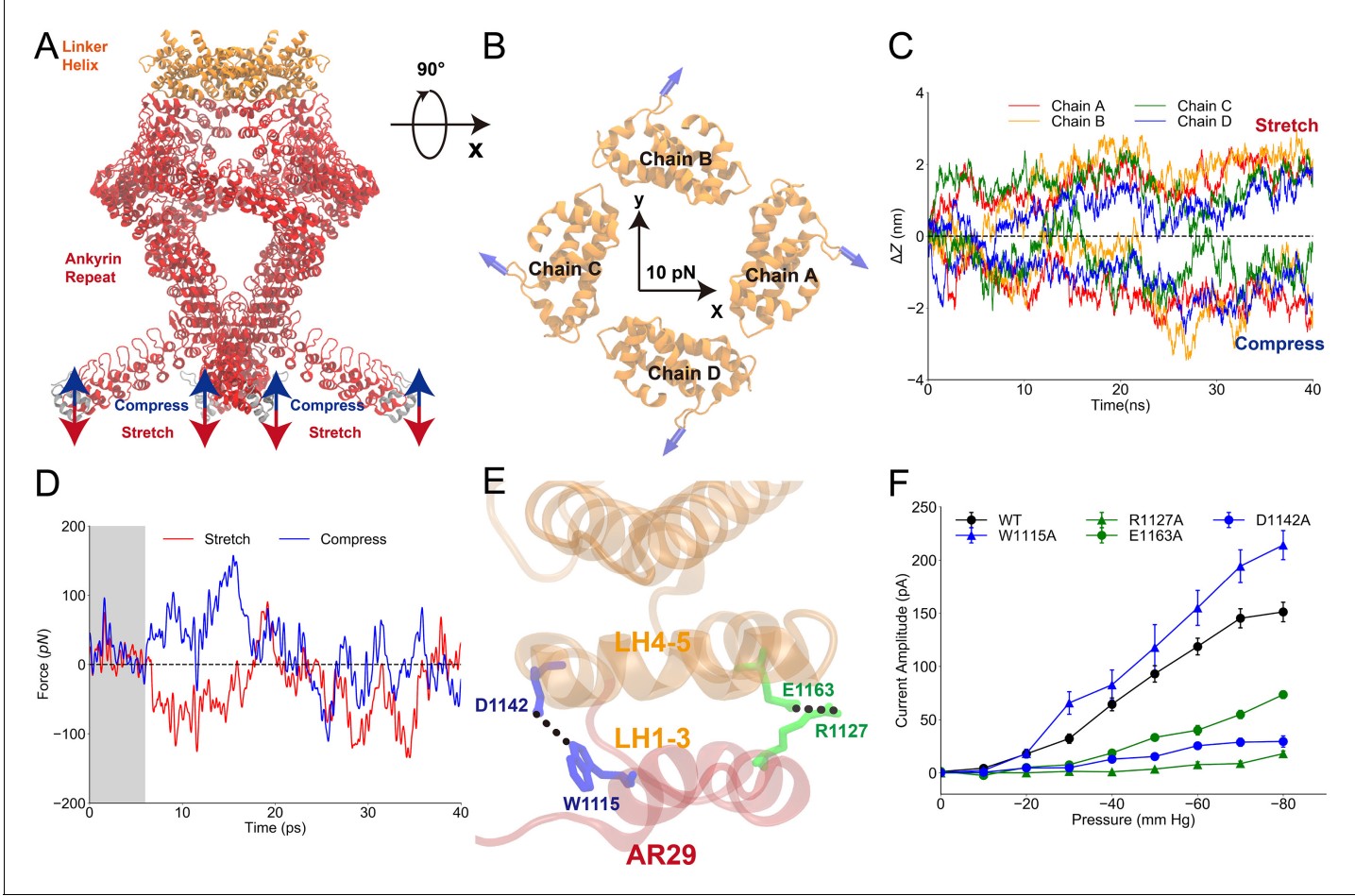

**Figure 3.** Mechanical properties of the ankyrin repeat (AR) region. (**A**) The simulation system in which the linker helix (LH) domain (orange) was restrained and a pushing or pulling force was applied to the first AR (gray). (**B**) Projection of the reaction forces of the restraints on the LH domain (same as the forces exerted on the transient receptor potential [TRP] domain by the LH domain) on the plane parallel to the membrane surface, showing that a torque is generated that will drive the LH and TRP domain to rotate clockwise (looking from the intracellular side). The calculation was based on the molecular dynamics (MD) trajectories CII1 and symmetrized from the original data as shown in *Figure 3—figure supplement 1*. (**C**) The AR region was compressed/stretched by a pushing/pulling force of 5 pN and reached its equilibrium length within 40 ns simulations (from MD trajectories SII1 and CII1 with respect to FII1). (**D**) The evolution of the net average reaction forces of the restraints on the LH domain when pushing (blue) or pulling forces (red) were applied to AR1, calculated from the MD trajectories FII2-6, CII2-6, and SII2-6 (*Supplementary file 1b*). A clear deviation occurred at around 7-8 ps during the simulation time, indicating that the forces applied to AR1 have reached LH at the time. (**E**) The residues forming two stable hydrogen bonds between the LH domain and AR29. (**F**) The mean and standard deviation (SD) of the mechano-gated current of the wild-type NompC and the mutants W1115A, D1142A, R1127A, and E1163A, under negative pressure in the outside-out patch-clamp experiments (wild type: n = 13; W1115A: n = 5; D1142A: n = 4; R1127A: n = 5; E1163A: n = 6). All of the error bars denote ± SD. Hydrogen bonds are indicated by dashed lines (**E**).

The online version of this article includes the following figure supplement(s) for figure 3:

**Figure supplement 1.** The forces exerted on the linker helix (LH) domain when AR1 was being pushed/pulled.

**Figure supplement 2.** Steered molecular dynamics (MD) of the single ankyrin repeat (AR) chain of NompC.

**Figure supplement 3.** The force constant of one ankyrin repeat (AR) chain in the AR bundle calculated from the molecular dynamics (MD) simulations with various pulling or pushing forces.

**Figure supplement 4.** The reaction forces to the restraints on the linker helix (LH) domain after applying a force on the AR1.

**Figure supplement 5.** The mutants of residues listed in *Figure 3F* showed normal membrane targeting.

**Figure supplement 6.** Formation of the alternative hydrogen bonds in the mutant, as identified in the molecular dynamics (MD) simulations (from chain A in the trajectories W1115A in *Supplementary file 1a*).

**Figure supplement 7.** Formation of the two hydrogen bonds that were not observed in the cryo-EM structure.

shown by the dashed line in *Figure 2F*. In addition, the mutations S1577A and R1581A on the TRP

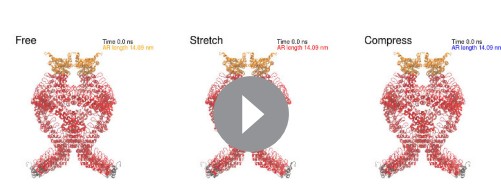

**Video 5.** The conformational changes of the AR domain in the "free", "pulling", and "pushing" MD simulations as shown in *Figure 3C*.
https://elifesciences.org/articles/58388#video5

domain, and K1244A and Q1253A on the LH domain, all resulted in significant loss-of-function, indicating the essential roles of these residues in conveying the forces from the AR region to the TRP domain. Thus, five out of seven residues, whose side chains form hydrogen bonds between the TRP and LH domains bonds as identified in our MD simulations, were crucial for the proper gating behavior of the NompC channel. The other two residues, D1236 and E1571, which are also involved in the hydrogen bonding between the TRP and LH domains, were found to be replaceable by adjacent residues in stabilizing the local conformation (*Figure 2—figure supplement 6*). These data show that W1572 acts as a rotation pivot by interacting with the S4-S5 linker, while the TRP domain senses a pushing force from the LHs upon AR compression. The force was stabilized by at least four hydrogen bonds, resulting in a clockwise rotation of the TRP domain around W1572. This is consistent with previous findings that the TRP domain (*Jin et al., 2017*; *Gao et al., 2016*), as well as the S4-S5 linker (*Jin et al., 2017*; *Cox et al., 2019*), play crucial roles in the gating of TRP or mechanosensitive channels. These results also confirm that a force/conformational change has to be transferred from the AR region to the TRP domain through the LHs when the NompC channel is opening in response to a mechanical stimulus.

## Mechanical properties of the AR region

To study how a pushing/pulling force is transferred to the LHs from the ARs, we performed multiple MD simulations on the truncated LH and AR domains (*Figure 3A*). We applied position restraints on the LHs (orange) and ran simulations with, or without, external forces applied to the terminal AR1 (*Figure 3A*). Several mechanical properties were obtained from these simulations. First, we analyzed the reaction forces of the position restraints on the LHs, which were identical, in magnitude and direction, to the forces acting on the LHs by ARs. The analysis indicated that when pushing the four AR1 toward the membrane with a total force of 20 pN (5 pN of force on each chain), the four AR29 apply a total torque of ~13 pN·nm on the LH domain pointing to the extracellular side, in addition to a dominant pushing force. This torque would help to rotate the TRP domain clockwise and drive the channel to open (*Figure 3B*, *Figure 3—figure supplement 1*). This is consistent with a continuum mechanics study by *Argudo et al., 2019*. Second, we calculated the force constant of the AR spring by $k = \frac{F}{z}$, where $F$ is the force we applied on AR1 and (*Sotomayor et al., 2005*) $z$ is the distortion of the AR region (*Figure 3C*, *Videos 5* and *6*). The spring constant of each AR helix was estimated to be $3.3 \pm 0.9$ pN nm$^{-1}$ in the supercoiled helix bundle formed by the four AR chains. Thus, the whole AR helix bundle has a force constant of ~13 pN nm$^{-1}$. For comparison, previous atomic force microscopy measurements determined a force constant of $1.87 \pm 0.31$ pN nm$^{-1}$ for a single AR chain (*Lee et al., 2006*), and previous steered MD (SMD) simulations obtained a value of ~4.0 pN nm$^{-1}$ (*Sotomayor et al., 2005*). However, our calculation was performed for the supercoiled AR helix complex, while the previous study evaluated a single 24-AR spring. For comparison, we performed SMD on a single 29-AR spring and estimated the spring constant to be $2.5 \pm 0.4$ pN nm$^{-1}$ (*Figure 3—figure supplement 2*). The close agreement of the values from the single AR and one AR in the supercoiled complex indicated that the four AR helices are not tightly coupled. We also performed simulations with weaker mechanical forces, ranging from 1 to 4 pN, and the resulting average force constants were all ~3 pN nm$^{-1}$ for each AR chain in the helix bundle (*Figure 3—figure supplement 3*). Therefore, the

**Video 6.** The conformational changes of the AR domain in the "free", "pulling", and "pushing" MD simulations, in which a 2-pN force was applied to each AR chain.
https://elifesciences.org/articles/58388#video6

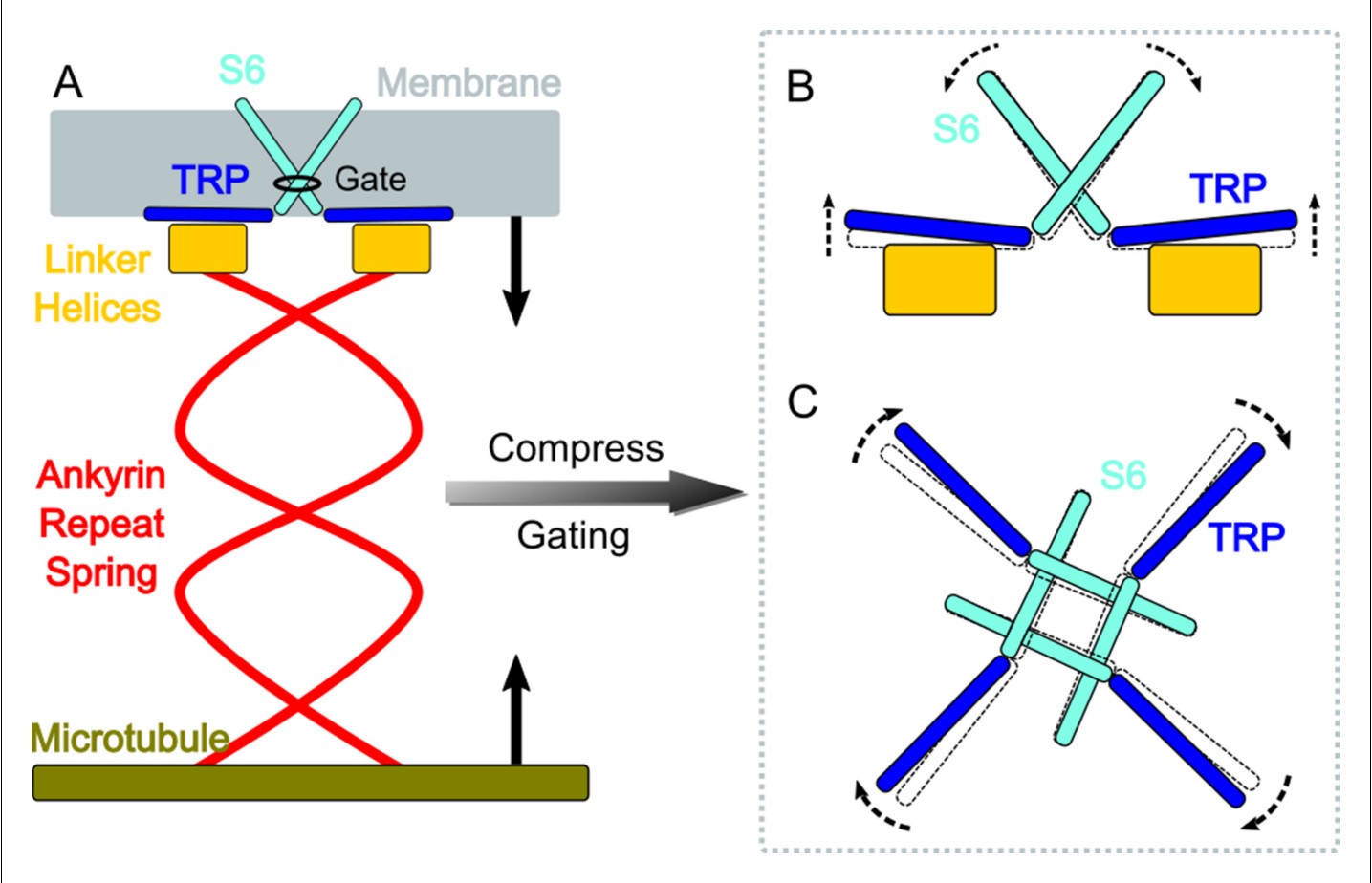

**Figure 4.** A gating model of NompC. (A) The compression of the ankyrin repeat (AR) region will generate a pushing force and a torque on the linker helix (LH) domain, pointing to the extracellular side. (B) The LH domain further pushes the transient receptor potential (TRP) domain, leading to a tilt (side view), and (C) a clockwise rotation of the TRP domain (looking from the intracellular side). The motion of the TRP domain pulls the S6 helices to slightly tilt and rotate, which dilates the constriction site of the pore.

AR supercoiled spring remained in its elastic limits in the study, although in the SMD trajectories with larger forces, the ARs showed a degree of distortion (*Video 5*). Third, we analyzed how fast the forces can be transferred from the AR1 to the LH. The deviation of the directions of the forces exerted on the LH regions when the AR region was stretched or compressed occurred after about 7-8 ps (*Figure 3D*, *Figure 3—figure supplement 4*). Considering that the length of the relaxed AR region is about 15 nm, we estimated that the force was transferred through the AR region at a speed of $1.8 \pm 0.2$ nm ps$^{-1}$. A recent study showed that forces are propagated via membranes at a speed of $1.4 \pm 0.5$ nm ps$^{-1}$ (*Aponte-Santamaría et al., 2017*). Therefore, it appears that the force transfer speed in the tethered NompC channel is comparable to, or slightly faster, than that in the membranes.

## Key residues at the interface between the AR and LH regions

We also found two stable hydrogen bonds between the ARs and LHs in the MD trajectories, between W1115 and D1142, and R1127 and E1163, respectively (*Figure 3E* and *Supplementary file 1d*). Mutations of D1142A, R1127A, and E1163A, which break the hydrogen bonds but show normal membrane targeting, led to a significant loss-of-function in the electrophysiology experiment (*Figure 3F*, *Figure 3—figure supplement 5*). However, W1115A does little to alter the mechano-sensing behavior (*Figure 3F*, *Figure 3—figure supplement 5*), probably because its hydrogen bonding and stabilizing role can be replaced by the adjacent Y1109, which can form a stable hydro-gen bond with D1142 in the presence of the W1115A mutation as observed in our MD simulations

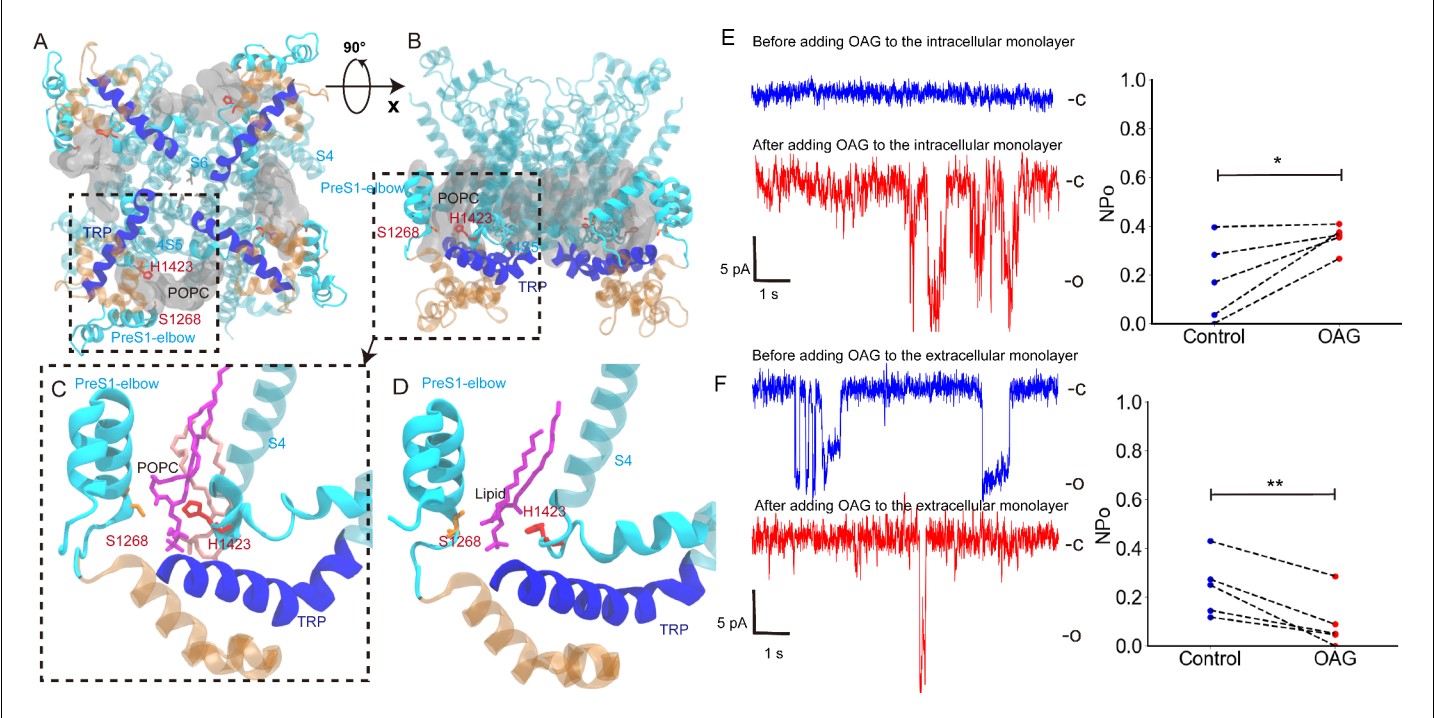

**Figure 5.** The interaction between H1423 and lipids and the effect of adding 1-oleoyl-2-acetyl-*sn*-glycerol (OAG) on the NompC opening. (**A**) The bottom view and (**B**) the side view of 1-palmitoyl-2-oleoyl-*sn*-glycero-3-phosphocholine (POPC) molecules moving around H1423. The transient receptor potential (TRP) domain is shown in blue, the S4S5 linker is shown in cyan, and the POPC density around H1423 is shown with silver transparent surfaces. The analysis was performed on the molecular dynamics (MD) trajectory FI0. (**C**) The initial (salmon) and final (violet) locations of a POPC molecule in the MD simulation trajectory FI0. (**D**) A lipid molecule was observed in the cryo-EM structure of NompC (PDB ID: 5vkq). The pocket between H1423 and S1268 can stably accommodate a lipid molecule in both the MD simulations and the cryo-EM structure. (**E**) The representative traces of the spontaneous NompC current before/after adding OAG to the intracellular monolayer of the membrane, and the corresponding average open probabilities (N = 5, paired Student's *t*-test, p=0.0208). (**F**) The representative traces of the spontaneous NompC current before/after adding OAG to the extracellular monolayer of the membrane, and the corresponding average open probabilities (N = 5, paired Student's *t*-test, p=0.0047).

(*Figure 3—figure supplement 6*). These results indicate that the interface between the AR and LH regions is crucial for the force transduction, further supporting the tethered spring model for NompC.

The hydrogen bonds between W1115-D1142 and R1127-E1163 were not observed in the cryo-EM structure. Our analysis showed that the distances between the side chains of the above two pairs of hydrogen bond-forming residues obtained from the MD trajectories were closer than those in the cryo-EM structure, while the distances between the α-carbon atoms were nearly identical (*Figure 3—figure supplement 7*). This indicated that the hydrogen bonding difference in the MD simulations and the cryo-EM structure was due to the side-chain adjustments in the simulations. As the resolution of the cryo-EM structure was insufficient to identify the exact locations of the side chains and the MD simulations accounted for all of the atomistic interaction details when dynamically evolving the systems, we believe that the MD simulations may have presented a better equilibrated local conformation that allowed identification of the two additional hydrogen bonds.

## Discussion

A combination study of MD simulations and electrophysiological experiments produced a clear 'push-to-open' gating model of the NompC channel. As illustrated in *Figure 4*, compression or shrinkage of cells can compress the AR spring, which generates a pushing force and also a torque on the TRP domain with a component pointing to the extracellular side and perpendicular to the membrane surface. The torque is generated by the specific supercoiled structure of the AR region, as demonstrated by the mechanics study of *Argudo et al., 2019*. This torque helps drive the TRP

domain to rotate clockwise. Our simulations indicate that the pushing force alone may be sufficient to generate a clockwise motion of the TRP domain, which in turn pulls the S6 helices to open the NompC gate. Critical residues between the TRP domain and LH, including R1581, W1572, Q1253, S1577, and K1244, as well as the presence of the S4-S5 linker above the TRP domain, ensure that the TRP domain will rotate clockwise around the pivot W1572 when a pushing force is applied to the LH region. This is consistent with a study showing that a TRPV1 mutant, which has only two ARs, can be mechanically opened by a pushing force (*Prager-Khoutorsky et al., 2014*). We believe that TRP channels similar to NompC, with a certain number of ARs, can be tethered to microtubules and use the push (-AR)-to-open mechanism to sense and respond to cell shrinkage or compression. This sensing mechanism can be complementary to the well-studied stretch(-membrane)-to-open mechanism that responds to cell expansion.

In this study, we focused on the intrinsic mechanical properties of NompC. To be more comprehensive, the possible effect of 'force-from-lipids' should be considered. Previous studies showed that lipid molecules may be involved in NompC gating and a stable lipid molecule has been found near H1423 in the cryo-EM map (*Jin et al., 2017*). Our analysis on MD simulation trajectories also showed that H1423 can stably interact with a 1-palmitoyl-2-oleoyl-*sn*-glycero-3-phosphocholine (POPC) lipid molecule. This lipid molecule was not in the interacting pocket in the initial simulation system but rapidly diffused to a location similar to that in the cryo-EM structure and acted as a bridge to link H1423 and S1268 (*Figure 5A–D*). This suggested that lipid molecules can help stabilize the local structure of NompC around H1423 and/or convey forces from the membrane.

A pushing/pulling force would lead to membrane curvature and a resulting asymmetrical TM stress profile that might activate mechanosensitive channels as demonstrated by *Cox et al., 2019*; *Bavi et al., 2016* Also, previous studies showed that the addition of OAG tends to activate TRPC6 by introducing an asymmetrical TM stress profile (*Spassova et al., 2006*; *Nikolaev et al., 2019*). Therefore, we conducted additional patch-clamping recordings to examine whether NompC is sensitive to the asymmetrical TM stress profile created by OAG. The experiments showed that the addition of OAG to the intracellular monolayer tends to activate NompC, while adding OAG to the extracellular monolayer had the opposite effect (*Figure 5E–F*). This finding was consistent with previous studies on TRPC6 (*Nikolaev et al., 2019*). The unidirectional OAG-induced activation suggests a synergetic gating mechanism. Pushing the AR spring would lead to the intrinsic gating of NompC and could simultaneously generate a membrane curvature and an asymmetrical stress profile that may also facilitate the channel gating. The synergetic gating mechanism between the force-from-tether and force-from-lipids warrants future research.

Due to limitations of the simulation timescale, we were unable to observe the full gating process of the NompC. Therefore, the ion conductance in the MD simulations was smaller than the experimental results for the fully open state. Nonetheless, a clear and detailed opening trend of the channel in the presence of pushing forces was seen in the MD simulations and this was supported by electrophysiological experiments. The combination of the two methods provides a plausible 'push-to-open' gating model for tethered mechanosensitive ion channels.

# Materials and methods

### Key resources table

| Reagent type (species) or resource | Designation | Source or reference | Identifiers | Additional information |
|---|---|---|---|---|
| Cell line (D. mel) | Schneider 2 (S2) cells | CCTCC (China Center for Type Culture Collection) | Serial# GDC0138 | Cell species report and *Mycoplasma* contamination test reports provided |
| Antibody | Rabbit anti-αNOMPC-EC (polyclonoal) | Ref. (*Zhang et al., 2015*) | | Immunostaining dilution (1: 500), primary antibody |
| Antibody | Alexa Fluor 594 AffiniPure Donkey Anti Rabbit IgG(H + L) | Yeason | Cat# 34212ES60 | Immunostaining dilution (1: 100), secondary antibody |
| Recombinant DNA reagent | pActin-Gal4 (plasmid) | Ref. (*Yan et al., 2013*) | | Plasmid for driving Gal4 expression under actin promoter in S2 cells |

*Continued on next page*

*Continued*

| Reagent type (species) or resource | Designation | Source or reference | Identifiers | Additional information |
|---|---|---|---|---|
| Recombinant DNA reagent | pUAST-NOMPC-EGFP (plasmid) | Ref. (*Yan et al., 2013*) | | Plamid for Gal4-driven NompC expression in S2 cells |
| Recombinant DNA reagent | pUAST-NOMPC-EGFP (del-miniwhite,dm) (plasmid) | This paper | | Plamid for Gal4-driven WT NompC expression in S2 cells, no miniwhite sequence |
| Recombinant DNA reagent | pUAST-NOMPC (D1236A)-EGFP(dm) (plasmid) | This paper | | Contains *Drosophila* NOMPC CDS with alanine substitution on D1236 |
| Recombinant DNA reagent | pUAST-NOMPC (R1581A)-EGFP(dm) (plasmid) | This paper | | Contains *Drosophila* NOMPC CDS with alanine substitution on R1581 |
| Recombinant DNA reagent | pUAST-NOMPC (K1244A)-EGFP(dm) (plasmid) | This paper | | Contains *Drosophila* NOMPC CDS with alanine substitution on K1244 |
| Recombinant DNA reagent | pUAST-NOMPC (E1571A)-EGFP(dm) (plasmid) | This paper | | Contains *Drosophila* NOMPC CDS with alanine substitution on E1571 |
| Recombinant DNA reagent | pUAST-NOMPC (Q1253A)-EGFP(dm) (plasmid) | This paper | | Contains *Drosophila* NOMPC CDS with alanine substitution on Q1253 |
| Recombinant DNA reagent | pUAST-NOMPC(S1577A)-EGFP(dm) (plasmid) | This paper | | Contains *Drosophila* NOMPC CDS with alanine substitution on S1577 |
| Recombinant DNA reagent | pUAST-NOMPC(S1421A)-EGFP(dm) (plasmid) | This paper | | Contains *Drosophila* NOMPC CDS with alanine substitution on S1421 |
| Recombinant DNA reagent | pUAST-NOMPC(W1572A)-EGFP(dm) (plasmid) | This paper | | Contains *Drosophila* NOMPC CDS with alanine substitution on W1572 |
| Recombinant DNA reagent | pUAST-NOMPC(W1115A)-EGFP(dm) (plasmid) | This paper | | Contains *Drosophila* NOMPC CDS with alanine substitution on W1115 |
| Recombinant DNA reagent | pUAST-NOMPC(D1142A)-EGFP(dm) (plasmid) | This paper | | Contains *Drosophila* NOMPC CDS with alanine substitution on D1142 |
| Recombinant DNA reagent | pUAST-NOMPC(R1127A)-EGFP(dm) (plasmid) | This paper | | Contains *Drosophila* NOMPC CDS with alanine substitution on R1127 |
| Recombinant DNA reagent | pUAST-NOMPC(E1163A)-EGFP(dm) (plasmid) | This paper | | Contains *Drosophila* NOMPC CDS with alanine substitution on E1163 |
| Chemical compound, drug | 1-Oleoyl-2-acetyl-*sn*-glycerol (OAG) | Sigma-Aldrich | Cat# O6754 | DAG analogue |
| Chemical compound, drug | GdCl$_3$ | Sigma-Aldrich | Cat# 439770 | NOMPC blocker |
| Chemical compound, drug | Concanavalin A (Con A) | Sigma-Aldrich | Cat# C5275 | Cell adhesion |
| Chemical compound, drug | ClonExpress II One-step Cloning Kit | Vazyme | Serial# C112 | Site-directed mutagenesis |
| Chemical compound, drug | TransIT-Insect Transfection Reagent | Mirus | Cat# MIR 6100 | S2 cell transfection |

## The simulation systems

We adopted a 'divide-and-conquer' strategy for the MD simulations and simulated two systems separately. System I included the TM region, the LH region, and the AR 29 of NompC (refer to *Supplementary file 1e* and *Figure 1—figure supplement 1* for the details of the residue range). The PPM server was used to reorient the NompC structure to ensure that the TM domain of NompC was well located in a lipid bilayer (*Lomize et al., 2012*). The protein was embedded in a POPC bilayer and then solvated in a water box of 150 × 150 × 150 Å$^3$. CHARMM-GUI was used to generate the configuration and topology of the simulation system, as well as the parameter files with the CHARMM36m force field (*Wu et al., 2014*; *Lee et al., 2016*; *Jo et al., 2009*). There were 492 POPC

molecules, 72,000 water molecules, and sodium and chloride ions corresponding to a concentration of 150 mM in the setup, resulting in a system of 314,352 atoms in total.

System II included the LH domain and the AR domain of NompC (refer to *Supplementary file 1e* and *Figure 1—figure supplement 1* for residue range details). The protein was solvated in a water box of $200 \times 200 \times 200$ Å³. CHARMM-GUI was used to generate the configuration, topology, and parameter files with CHARMM36m force fields. In addition to the protein, 354,567 water molecules were added and sodium chloride ions were added to maintain an ion concentration of 150 mM. The simulation system II contained 1,134,213 atoms in total.

## MD simulations

All of the MD simulations were performed with GROMACS 5.1.2 (*Hess et al., 2008*). The REDUCE program in AMBER was used to add hydrogens to the original PDB files and determine the protonation state of the histidine residues (*Word et al., 1999*; *Case et al., 2005*). For system I, energy minimization was achieved using the steepest descent algorithm, followed by a two-stage equilibration, a 0.4 ns NVT (constant particle number, volume, and temperature) equilibration simulation with harmonic restraint applied to the protein molecules (a force constant of 4000 kJ mol$^{-1}$ nm$^{-2}$ on the backbone and 2000 kJ mol$^{-1}$ nm$^{-2}$ on the side chains), and a 20 ns NPT equilibration simulation with gradually decreased restraint (from 2000 to 100 kJ mol$^{-1}$ nm$^{-2}$ on the backbone and from 1000 to 50 kJ mol$^{-1}$ nm$^{-2}$ on the side chains). During the equilibration processes, harmonic restraints were applied to heavy atoms of the protein, and planar restraints were used to keep the positions of lipid head groups along the normal direction of the membranes. The simulation temperature of the system was set to 300 K. After all of the equilibration steps were completed, the restraints were removed and the production simulations were performed in the NPT ensemble. The time step was 2 fs. The cubic periodic boundary condition was used during the simulations and the van der Waals interaction was switched off from 10 to 12 Å. The long-range electrostatic interactions were calculated with the particle mesh Ewald (PME) method (*Darden et al., 1993*).

For system II, the steepest descent algorithm was used to achieve initial energy minimizations, and then it was followed by a two-stage equilibration, a 0.2 ns NVT equilibration simulation with harmonic restraint forces applied to the protein (force constants of 400 kJ mol$^{-1}$ nm$^{-2}$ on the backbone and 40 kJ mol$^{-1}$ nm$^{-2}$ on the side chains), and a 10 ns NPT equilibration simulation with restraints on the protein backbone (force constant of 400 kJ mol$^{-1}$ nm$^{-2}$) and side chains (force constant of 40 kJ mol$^{-1}$ nm$^{-2}$). The temperature was set to 300 K. In the production simulations of system II, 1000 kJ mol$^{-1}$ nm$^{-2}$ harmonic restraints were applied to the heavy atoms of the LH domain while the restraints on the AR region were removed. The time step was set to 2 fs, and the trajectories were saved every 10 ps. The long-range electrostatic interactions were calculated using the PME method (*Darden et al., 1993*).

## SMD simulations

For system I, after equilibration, SMD simulations were utilized to pull AR29 to simulate the mechanical stimuli from the AR region (spring) (*Izrailev, 1999*; *Isralewitz et al., 2001*). The TM regions of the four chains of NompC were treated as the reference group, and the AR29 of the four chains were treated as the pulling group. In addition to the force-free simulations (no pulling forces on the AR region), we considered the two most essential mechanical stimuli: the pulling and pushing forces on the AR29 along the direction normal to the membrane surface (the z-axis in our simulations), where pulling meaning that the force is pointing to the intracellular side (stretch of the AR spring) and pushing meaning that the force is pointing to the extracellular side (compression of the AR spring) along the z-axis. We tested a series of harmonic force constants as well as pulling speeds, and a harmonic force constant of 100 kJ mol$^{-1}$ nm$^{-2}$ and a pulling speed of 0.1 Å ns$^{-1}$ were found to be reasonable for the gating simulations, where the opening of the pore was observed and the global protein structure was not disrupted in 200 ns. As further validations, a series of weaker SMD simulations with smaller force constants of 50 kJ mol$^{-1}$ nm$^{-2}$ and a slower pulling speed of 0.05 Å ns$^{-1}$ were performed for each condition (*Supplementary file 1a*). During the MD/SMD simulations, the distances between the TM region of NompC and AR29, and the driving forces that act on the four chains of AR29 were recorded. The frames from MD/SMD trajectories were saved every 1 ns. All of the MD/SMD trajectories of system I are listed in *Supplementary file 1a*.

For system II, starting from the equilibrated structure, the LH domain was position-restrained and SMD simulations were performed to pull AR1, simulating the compression and stretch of the AR region (*Izrailev, 1999*; *Isralewitz et al., 2001*). The LH domains of the four chains of NompC were treated as the reference group, and the AR1 of the four chains were treated as the pulling groups. Constant pulling forces of 5 pN were applied on AR1 of each chain along the z-axis. Again, we considered the two most essential mechanical stimuli here: the pulling and pushing forces on the AR1, where pulling means the force is pointing to the intracellular side (a stretch of the AR spring), and pushing means the force is pointing to the extracellular side (compression of the AR spring) along the z-axis. In addition, we performed many 100 ns MD/SMD trajectories with a series of pushing/pulling forces ranging from 0 to 5 pN (*Supplementary file 1b*). During the MD/SMD simulations, the distances between the LH domain and the AR1 of each chain were recorded. To take into account the position restraints applied to the AR1 of the four chains by microtubules, an additional flat bottom potential of 100 kJ mol$^{-1}$ nm$^{-2}$ with a 3 nm radius was added on the four chains of AR1 on the x-y plane, to restrain each AR1 to move within a cylinder parallel to the z-axis. To estimate the mechanical property of a single chain of AR, the same protocol was applied to the single chain A of system II.

To estimate the speed of the forces that are conveyed along the AR spring, five 40 ps trajectories (FII2-6, CII2-6, and SII2-6 in *Supplementary file 1b*) were generated for each condition (free, compress/push, and stretch/pull) with a constant force of 5 pN on the AR1 of each chain. These trajectories were saved every 10 fs, a frequency high enough for the force transfer analysis. All of the MD/SMD trajectories of system II are listed in *Supplementary file 1b*.

## The ion permeation simulations

After 200 ns pushing SMD simulations for system I, the TM pore of NompC was partially opened (*Figure 1D*). This partially open state was then stimulated by an umbrella pushing potential with a force constant of 100 kJ mol$^{-1}$ nm$^{-2}$ and an initial force of 50 pN on the AR29 of each chain for more than 500 ns, and a transient structure with the pore radius of the lower constriction more than 2.0 Å was exacted at 545 ns for the ion permeation simulations. The gate region, which includes S5, S6, the selectivity filter, and the TRP region, was position-restrained by the harmonic potential with a force constant of 1000 kJ mol$^{-1}$ nm$^{-2}$, while a TM potential of 300 mV was applied by setting a uniform electric field along the z-direction. Three independent 200 ns MD trajectories were generated with 150 mM KCl or NaCl in the systems, respectively. All of the ion permeation simulations of the partially open NompC are listed in *Supplementary file 1c*. The ion permeation events were analyzed in the simulations, from which we calculated the current by $I = \Delta q / \Delta t$, which was then used to calculate the conductance by $C = I/U$, where $U$ was the TM potential 300 mV. The estimated conductance of the channel was about 7-15 pS.

## Mutation MD simulations

To determine why the single mutations of critical residues D1236A, E1571A, and W1115A did not significantly impact the gating of NompC, three 500 ns all-atom MD simulations were performed with the mutations D1236A, E1571A, and W1115A incorporated into the system, respectively (*Supplementary file 1a*).

## Principal component analysis

The distances between the centers of mass of the TM domain and AR29 (TM-AR29 distance) were monitored in all of the MD simulations for system I (*Figure 1—figure supplements 11–12*). The data after a sharp change of the distance were discarded for further analysis, where the global conformation was distorted due to the strong pulling forces (gray areas in *Figure 1—figure supplement 11*). On the other hand, the overlaid initial and 200 ns conformations from the MD trajectories FI0, SI0, and CI0 indicated that the global conformation remains undistorted before the sharp change of distance (*Figure 1—figure supplement 13*); 950 protein structures (500 frames from FI0, the first 250 frames from SI0, and the first 200 frames from CI0) were concatenated for the PCA.

## Analysis of the motion of the TRP domain

The motion of the TRP domain was characterized by the tilt angle and the rotation angle. The tilt angle of the TRP domain was defined as the variation of the angle between the z-axis and the axis of the TRP domain with respect to that of the initial conformation. The rotation angle was defined as the angle between the XY-plane projection of the axis of the TRP domain with that of the initial conformation.

## Analyzing the role of the AR region in the force convey

In the simulations of system II, we calculated the reaction forces of the restraint on the LH domain of each chain of the tetramer, whose average magnitude and direction should be the same as the forces exerted on LH by the ARs. After that, these forces were projected on the x-y plane and z-direction (*Figure 3—figure supplement 1*). The calculation was performed from 20 to 40 ns in the trajectories CII1 and SII1, and from 50 to 100 ns in the trajectories CII8-17 and SII8-17.

For the calculation of the force constant of the ARs, the distance between the center of the LH and AR1 (the length of the AR region) was monitored in the MD trajectories. After the length of the AR region became stable, the force constant was calculated by using the formula $k = F/\Delta z$, where $F$ was the applied force on each chain, and $\Delta z$ was the variation of the length of the AR region with respect to the same value in the force-free simulations.

To estimate the force transfer speed through the AR region, we analyzed how long it took for the force applied to the AR1 to impact the LH domain (*Figure 3—figure supplement 4*). We generated five short trajectories (40 ps each) for the free/pushing/pulling simulations of system II with high output frequency (10 fs per frame). In the first stage of each trajectory, the force on the LH region could not be distinguished by the simulation conditions, and then, at some point, the force values on the LH domain started to deviate among the free/pushing/pulling simulations (*Figure 3—figure supplement 4*, where gray areas end). This represents the forces applied to AR1 starting to impact the LH domain. With this, we estimated that the forces applied on AR1 required about 6-12 ps to arrive at the LH domain, and the speed of force transfer was estimated to be $1.8 \pm 0.2$ nm ps$^{-1}$ along the AR region.

## Cell lines

S2 cells were purchased from China Center for Type Culture Collection (CCTCC), Serial number: GDC0138. The S2 cell line authentication was confirmed by COI authenticate. The mycoplasma contamination was tested negative by fluorescence quantitative PCR. The above cell tests were performed by a third party, Jiangsu Micro Spectrum Detection Technology Co., Ltd, and the cell test report Number was WJS-21046354-HJ-01-ER1.

## Electrophysiological recording

*Drosophila* S2 cells were cultured in Schneider's Insect medium supplied with 10% FBS at 27°C. TransIT-Insect Transfection Reagent (Mirus) was used to transfect cells according to the product protocol. The miniwhite region was deleted from all of the pUAST-NompC-EGFP plasmids (*Supplementary file 1f* and *Figure 2—figure supplement 4*) to enable site-directed mutagenesis. pUAST-NompC-EGFP (wild-type or mutants) constructs were co-transfected with pGal4 (*Yan et al., 2013*). Recordings were carried out 36–48 hr after transfection. Cells were transferred onto glass slides, pre-coated with Con A 30 minutes prior to recording.

Electrophysiological recordings were conducted under an Olympus CKX41 microscope equipped with a 40× water immersion lens. Transfected cells were identified by green fluorescence. The sample rate was 10 kHz and filtered at 1 kHz (low-pass). Patch electrodes with 12–20 MΩ resistance were used. The bath solution contained 140 mM NaMES (sodium methanesulfonate) and 10 mM HEPES. For cell-attach mode recording, the pipette solution was the same as the bath solution. For inside-out and outside-out mode recording, the pipette solution contained 140 mM potassium D-gluconate (CsMES in OAG application experiment) and 10 mM HEPES. All of the solutions were adjusted to 285 mOsm and pH 7.2.

After forming a specific recording mode (cell-attach mode, inside-out mode, or outside-out mode), negative pressure or positive pressure was applied to the excised membrane via a high-

speed pressure clamp (HSPC, ALA-Scientific). Signals generated from pClamp software were sent to HSPC to control the timing and intensity of the pressure.

To record the dose-response curve of the mechanosensitive current, pressure steps of 500 ms with 10 mm Hg increment (for inside-out and outside-out recording) or 20 mm Hg increment (for cell-attach recording) were applied to the membrane patch through the recording pipette. The inside-out and outside-out patch-clamp traces under different pressure are shown in *Figure 1—figure supplement 9A* while the mean currents under different pressure are shown in *Figure 1—figure supplement 9B*.

## Mutation generation

All of the point mutations on pUAST-NompC-EGFP plasmid were introduced by site-directed mutagenesis using a CloneExpress II One-step Cloning kit (Vazyme) and confirmed via sequencing of the mutation region. Further experiments were performed the same as outside-out and inside-out patch clamp in the wild-type NompC described in the electrophysiological recording.

## Non-permeablized immunostaining of membranous NompC

For non-permeablized staining, the transfected cells were fixed and incubated with 4% paraformaldehyde at room temperature for 10 min. The cells were then washed with PBS three times and blocked with 4% BSA at 37°C for 100 min. The primary antibody (rabbit anti-αNOMPC-EC, 1:500; used in *Zhang et al., 2015*) was diluted in PBS and incubated with transfected cells at 4°C overnight. Cells were then washed with PBS three times and incubated with secondary antibody (Alexa Fluor 594 AffiniPure Donkey Anti-Rabbit IgG(H + L), 34212ES60, Yeasen) for 90 min at room temperature. After being washed briefly with PBS, cells were mounted on a coverslip for imaging.

## Drug application

$GdCl_3$ was dissolved in the bath solution (both bath and electrode solutions under cell-attached mode) to a final concentration of 100 µM. OAG was dissolved in DMSO and diluted in bath solution to a final concentration of 30 µM. The time window of 10 s was used to calculate the spontaneous open probability of NompC ($NP_0$).

## Acknowledgements

We thank Prof Wei Zhang at Tsinghua University for his generous sharing the antibody with us. The research was supported by the National Natural Science Foundation of China (32071251 and 21873006 to CS; 31571083 and 31970931 to ZY), and the National Key Research and Development Program of the Ministry of Science and Technology of China (2016YFA0500401 to CS; 2017YFA0103900 and 2016YFA0502800 to ZY). ZY was supported by the Program for Professor of Special Appointment (Eastern Scholar of Shanghai, TP2014008), the Shanghai Municipal Science and Technology Major Project (No. 2017SHZDZX01) and ZJLab, and the Shanghai Rising-Star Program (14QA1400800). Part of the MD simulation was performed on the Computing Platform of the Center for Life Sciences at Peking University, and part of the MD simulation was performed on the Tianhe II supercomputer in the National Supercomputing Center in Tianjin.

## Additional information

### Funding

| Funder | Grant reference number | Author |
|---|---|---|
| National Natural Science Foundation of China | 32071251 | Chen Song |
| National Natural Science Foundation of China | 21873006 | Chen Song |
| National Natural Science Foundation of China | 31571083 | Zhiqiang Yan |
| National Natural Science | 31970931 | Zhiqiang Yan |

| Foundation of China | | |
| --- | --- | --- |
| Ministry of Science and Technology of the People's Republic of China | 2016YFA0500401 | Chen Song |
| Ministry of Science and Technology of the People's Republic of China | 2017YFA0103900 | Zhiqiang Yan |
| Ministry of Science and Technology of the People's Republic of China | 2016YFA0502800 | Zhiqiang Yan |
| Program for Professor of Special Appointment, Eastern Scholar of Shanghai | TP2014008 | Zhiqiang Yan |
| Shanghai Municipal Science and Technology Major Project | No.2017SHZDZX01 | Zhiqiang Yan |
| Shanghai Rising-Star Program | 14QA1400800 | Zhiqiang Yan |

The funders had no role in study design, data collection and interpretation, or the decision to submit the work for publication.

### Author contributions

Yang Wang, Data curation, Formal analysis, Validation, Investigation, Visualization, Writing - original draft, Writing - review and editing; Yifeng Guo, Data curation, Formal analysis, Validation, Investigation, Writing - original draft, Writing - review and editing; Guanluan Li, Data curation, Validation; Chunhong Liu, Formal analysis, Validation, Writing - review and editing; Lei Wang, Formal analysis, Investigation; Aihua Zhang, Software, Formal analysis, Validation; Zhiqiang Yan, Conceptualization, Resources, Supervision, Funding acquisition, Validation, Investigation, Methodology, Project administration, Writing - review and editing; Chen Song, Conceptualization, Resources, Supervision, Funding acquisition, Investigation, Visualization, Methodology, Writing - original draft, Project administration, Writing - review and editing

### Author ORCIDs

Yang Wang (iD) https://orcid.org/0000-0002-5504-9800
Chunhong Liu (iD) http://orcid.org/0000-0002-3666-2343
Chen Song (iD) https://orcid.org/0000-0001-9730-3216

### Decision letter and Author response

Decision letter https://doi.org/10.7554/eLife.58388.sa1
Author response https://doi.org/10.7554/eLife.58388.sa2

## Additional files

### Supplementary files

• Supplementary file 1. Tables. (a) Molecular dynamics/steered molecular dynamics (MD/SMD) trajectories of system I. (b) MD/SMD trajectories of system II. (c) Ion permeation simulations of the partially opened NompC. (d) Stable hydrogen bonds and their occupancies in the MD/SMD trajectories. (e) Different domains/components used in the MD simulations. (f) The primers used for the alanine substitution.

• Transparent reporting form

### Data availability

All data generated or analysed during this study are included in the manuscript and supporting files. Numeric data files have been provided for Figure 1 B C D F G, Figure 1—figure supplements 2, 3, 4, 5, 8, 9, 10, 11, 12, Figure 2 A F, Figure 2—figure supplements 1, 2, 5, 6, Figure 3 C D F, Figure 3—figure supplements 1, 2, 3, 4, 5, 6, 7 and Figure 5 E, F.

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
