## [Decision Letter]

**Acceptance summary:**

The manuscript by Wang et al. describes a worthwhile study proposing a novel mechanism for the NompC ion channel gating by mechanical stimuli, demonstrating that NompC can only be open by application of mechanical force by pushing the bundle of ankyrin repeats outwards, causing the clockwise rotation of the pore helices and thus opening the channel. The model is interesting and appealing and given the rapid pace at which the mechanotransduction field is currently growing, the reviewers judge that this study will be of interest to a broad readership interested in molecular mechanobiology.

**Decision letter after peer review:**

Thank you for submitting your article "The push to open mechanism of the tethered mechanosensitive Ion channel NompC" for consideration by *eLife*. Your article has been reviewed by 3 peer reviewers, and the evaluation has been overseen by a Reviewing Editor and Richard Aldrich as the Senior Editor. The following individuals involved in review of your submission have agreed to reveal their identity: Boris Martinac (Reviewer #1); Rachelle Gaudet (Reviewer #3).

The reviewers have discussed the reviews with one another and the Reviewing Editor has drafted this decision to help you prepare a revised submission.

As the editors have judged that your manuscript is of potential interest, but as described below that several additional experiments and simulations are required before the conclusions' credibility can be assessed, we would like to draw your attention to changes in our revision policy that we have made in response to COVID-19 (https://elifesciences.org/articles/57162). First, because many researchers have temporarily lost access to the labs, we will give authors as much time as they need to submit revised manuscripts. We are also offering, if you choose, to post the manuscript to bioRxiv (if it is not already there) along with this decision letter and a formal designation that the manuscript is "in revision at *eLife*". Please let us know if you would like to pursue this option. (If your work is more suitable for medRxiv, you will need to post the preprint yourself, as the mechanisms for us to do so are still in development.)

Summary:

The manuscript by Wang et al. describes a worthwhile study proposing a novel mechanism for the NompC ion channel gating by mechanical stimuli. NompC has to date been the only TRP-type ion channel that has convincingly been shown to be gated by the tethered mechanism. In this study, the authors used a combination of molecular dynamics simulations and patch clamp recording to demonstrate that NompC can only be open by application of mechanical force by pushing the bundle of ankyrin repeats outwards (with respect to the cell interior). This is consistent with what the 'rough' model of opening that Argudo et al. published in 2017 based on a combination of their continuum mechanics calculations and comparisons to the open state of the TRPV1 channel structure. The NompC ankyrin repeat bundle is proposed to act as a spring, which when pushed (compression of the spring) should cause the clockwise rotation of the pore helices and thus opening the channel. Pulling on the spring (stretching the spring) has a negligible effect on the channel opening. Simulation results are complemented by electrophysiology experiments in *Drosophila* S2 cells carried out to show that positive pressure (corresponding to compression) causes channel gating, but that much less current is evoked from negative compression. The model suggested is interesting and appealing and given the rapid pace at which the mechanotransduction field is currently growing, the reviewers judges that this study will be of interest to a broad readership interested in molecular mechanobiology.

There are, however, significant scientific issues and issues of presentation that made the manuscript as presented challenging to evaluate. The reviewers have therefore listed several aspects that need to be revised before the paper can be accepted for publication.

Essential revisions:

The majority of the conclusion stem from the interpretation of the MD simulations. The reviewers, however, expressed concern over the length and small number of repeats. Indeed, the simulations are very short for such large deformations, and the reviewers were unsure if it was possible to extract the true behavior of protein to a mechanical force on the 100 ns time scale, when the true force is applied over the microsecond to millisecond timescale. It is noteworthy that mechanical responses of proteins tend to have a huge dependence on time scale. Given the system size, the reviewers judge that it should be easy enough to double the time scale and collect many replicates to ensure that the results are robust. We note that given the issues related to presentation (see below), it is possible that the reviewers have misinterpreted the results and repeats were indeed carried out. For example, there were simulations of System I with different force constants, but those don't seem to be discussed anywhere in the text. Related to the interpretation of the MD simulations, the reviewers had the following comments:

The conformational changes observed in Figure 2b,c (and Supplemental Video S1C) are very small (especially panel c). Does this really open the channel? More analysis (with ions moving through the constriction zone for instance or water wetting/dewetting changes) must be done to support this claim. How much water is seen moving back and forth? What about free ion movement into and out of the cavity. Is unbiased ion entry to the cavity observed? Moreover, what about the local pore structure in the pushing experiment shown in Supp Video S1C really starts to open, is it side chain movement or all backbone? What are the backbone radius changes?

Figure 2d,e – is this based on gazing at the structure itself, or did were these residues identified from changes in the simulations? The authors mention that the values come from the simulations, but one wonders if they could have been guessed from the structure themselves.

Similarly, are Figure 3e residues identified from simulation or sequence gazing at the structure? Additionally, the authors state 'we found two stable hydrogen bonds between the ARs and LHs in our MD trajectories, between W1115 and D1142, and R1127 and E1163, respectively'. Cheng and Jan did not find potential hydrogen bonds directly from the atomic model of NOMPC at these two sites structure. The two pairs are either too far to establish a hydrogen bond (D1142-W1115: 5.4 Å) or on the borderline (E1163-R1127: 3.9 Å). The authors should comment on this point – perhaps this points to an important role in the simulations, but why do the simulations show h-bonding while the structure does not?

Looking at the Video S5: in all three simulations illustrated, the AR3-AR4 packing is disrupted. This is not something expected, yet it is not addressed anywhere in the manuscript and should be addressed. This will affect how the forces are transmitted through the protein.

The simulations of System II seem to suggest that pushing and pulling on AR1 while restraining the LH domain lead to the LH domain experiencing rotational forces (e.g. Figures 3b and S7). It seems like these are the forces that should be applied to System I, rather than pushing or pulling perpendicularly to the membrane plane.

The authors attempt to provide some experimental evidence that NompC is only pushed open by showing NompC failed to respond to negative pressure using cell-attached patch-clamp. However, previously the Jan lab performed the same experiment (Figure 5F, Zhang et al., Cell 2015), and their data showed NompC in fact could respond to negative pressure. The authors need to explain the inconsistency between their results and what the Jan lab has shown.

The authors state on p. 5, l. 10-102 that "Previous studies have shown that the AR regions are necessary for the mechano-gating of NompC, and the force from lipids cannot open the channel in the absence of the AR region (15)." However, this statement is incorrect: on p. 1401 of ref. 15, the authors of the Zhang et al., 2015 write: "We wish to emphasize that while our results strongly support the notion that ARs function as a tether for mechanogating of NOMPC, our results do not exclude the potential role of interactions between NOMPC protein and the membrane lipids nearby." And indeed, in the 2017 Nature paper by Jin et al. there is evidence that His1423 in the S4-S5 linker of NompC coordinates the lipid polar head group and is required for the channel mechanosensitivity because "the His1423Ala mutant is no longer responsive to mechanical stimulation" (p. 2). And further on p. 4 of this paper: "While it is unlikely that membrane deformation alone is the driving force in NOMPC gating, as in the case of other mechanosensitive channels such as Piezo and TRAAK channels, the proximity of a lipid molecule interacting with the functionally important His1423 at the S4-S5 linker suggests that lipid-protein interactions may have an important role in channel activity." The question is how to reconcile the views of Zhang et al., Jin et al. and the authors of this study, who seem to exclude the possibility of the bilayer involvement in NompC mechanogating? To answer this question, the reviewers have the following suggestion, which can be tested experimentally (and possibly using MD as well) by the authors of this study. Given the unidirectionality of the force causing the channel to open, it is likely that pushing the ankyrin bundle spring would locally curve the membrane bilayer. The local membrane curvature can activate a mechanosensitive channel by changing the asymmetry of the transbilayer pressure profile favouring the channel opening (Cox et al., Cell Reports 29(1): 1-12, 2019). In contrast, the force pulling on the spring would change the local membrane curvature in the opposite direction favouring the closed conformation of the channel. Given that the lipid bilayer is much more compliant compared to a membrane protein the displacement of the NompC channel with respect to the lipid bilayer by pushing the ankyrin spring could trigger the channel open conformation (Kung, Nature 436: 647-654, 2005). As shown by FE modelling (Bavi et al., Membranes 6: 14, 2016) a local curvature corresponding to a radius of less than 50 nm results in membrane tension of several mN/m, which is sufficient to open a mechanosensitive channel. The unidirectionality of the activation force reflecting the pushing vs. pulling the spring in the NompC case can be tested experimentally by inserting an amphipathic molecule, such as OAG (1-oleoyl-2-acetyl-*sn*-glycerol), a commercially available diacylglycerol (DAG) analogue, which like other amphipathic molecules can bend the cell membrane and was shown to activate the TRPC6 channel, which could not be activated by membrane stretch (Nikolaev et al. (2019) J Cell Sci 132: jcs238360). By adding OAG into the extracellular monolayer of the membrane bilayer it may be possible to open NompC (corresponding to pushing force), whereas adding it to the intracellular monolayer (corresponding to pulling force) would not open the channel. Thus the request from the reviewers is to perform patch clamp experiments using inside-out patch configuration as described in their manuscript. By adding OAG to the extracellular monolayer of the membrane patch accessible from the bath they should be able to open the NompC channel, whereas adding OAG from the pipette they would not be able to do it. Provided the authors could demonstrate this effect of OAG on NompC gating, these experiments together with the results of their study described in this manuscript would reconcile the tethered mechanism with the force-from-lipids paradigm in the case of NompC and indicate that "sharing the force" between the membrane bilayer and cytoskeleton is the likely mechanism of NompC activation by mechanical force.

Relatedly, P. 10, l. 205-206: "These results indicate that the interface between the AR and LH regions is also crucial for the force transduction, further confirming the tethered spring model for NompC." – Further to this statement the authors omitted to mention that His1423 residue within the S4-S5 linker was reported to be functionally important for NompC mechanosensitivity.

On P. 6, l. 125-128, the authors write: "Thus, consistent with previous structural studies of TRPV1.… it is the pushing force (compression of the intracellular domain) that leads to such a collective gating motion." – This is correct. However, it does not completely explain the mechanism of opening the channel. As stated above, the force opening the channel can also result from a change in the local membrane curvature causing the channel to open given that the force pushing on the ankyrin bundle spring could generate positive curvature and consequently change asymmetry of the transbilayer pressure profile as indicated by the FE modelling study (Bavi et al., Membranes 6: 14, 2016).

Additionally, the reviewers judged that many of the conclusions are over-interpreted:

Many of the mutations resulted in non-responsive channels. These non-responsive channels could be non-responsive for many reasons, not only gating defects (e.g. folding defects). The authors need to show that the membrane localization and quantity of NompC at the membrane is not affected by the mutations at the least.

The ion conductance in the simulation being lower than experimentally measured is interpreted as the channel conformation being only "partially open". But many other reasons could explain the observed conductance. Are there properties of these currents that you can extract that are not dependent on current or, more importantly, can single channel recordings be performed to confirm the impact on channel function?

Hypotheses about compensatory interactions in the hydrogen-bonding networks, etc, are presented as conclusions.

P. 9, L. 191-196: "Considering the fact that the length of the relaxed AR region is about 15 nm, we estimated that the force was transferred through the AR region at a speed of 1.8 ± 0.2 nm/ps.… the force is propagated via the membrane at a speed of 1.4 ± 0.5 nm/ps (37). Therefore, it appears that the force transfer speed in the tethered NompC channel is faster than that in the membranes." This statement requires statistical testing to compare quantitatively the two speeds of force propagation. As presented here, the force propagation through AR region does not appear to be significantly faster from the force propagation via the membrane bilayer. In contrary, they seem quite comparable.

P. 11, l. 232-234: "Nonetheless, a clear and detailed opening process of the channel in the presence of pushing forces was revealed by our MD simulations and validated by electrophysiological experiments." – MD simulations have not been validated experimentally because in outside-out patches CSK connections to the membrane bilayer/channels are largely disrupted, which has been shown to happen upon gigaohm seal formation (see for example Suchyna et al., Biophys J. 97(3): 738-747, 2009). Consequently, opening NompC by pushing the ankyrin spring is not the likely mechanism of the channel opening in membrane patches as suggested by the model shown in Figure 4A of the manuscript.

The reviewers identified presentation issues that obscured the message of the paper:

The methods and Table S1 indicate that three separate simulations with three different force constants were performed. But the authors do not indicate which of these simulations they used in each the figures or analyses. This lack of clarity as to which simulation is used in the analyses is pervasive throughout the manuscript, making a lot of the data difficult to interpret and evaluate. In general, all figure legends or specific mentions of simulations in the main or supplemental text should include a reference to the trajectory labels provided in Tables S1-S3.

The concatenation for PCA analysis is fine, but the data should be "unconcatenated" clearly in the figures. This is done in Figure 2A (although using saturation gradients to indicate progression through each of the three simulations would be useful). I.e. in Figure S3, for example, the three different simulations should clearly be demarcated.

The methods section reads more like a mixture of results and methods. The manuscript should be reorganized to separate the two. The methods are also inadequately detailed: Residue numbers should be used to indicate the residues included in various simulations, residues restrained in simulations, residue ranges used as AR1, AR29, LH, TM, etc. The published structure, 5VKQ, is missing residues 1603-1669 between the TRP box and the C-terminal helix that completes the LH domain, how was this handled in building the simulated systems?

The methods should provide enough detail that the exact full sequence of the expression vectors (including the codons used for the mutations) can be reconstructed.

---

## [Author Response]

Essential revisions:The majority of the conclusion stem from the interpretation of the MD simulations. The reviewers, however, expressed concern over the length and small number of repeats. Indeed, the simulations are very short for such large deformations, and the reviewers were unsure if it was possible to extract the true behavior of protein to a mechanical force on the 100 ns time scale, when the true force is applied over the microsecond to millisecond timescale. It is noteworthy that mechanical responses of proteins tend to have a huge dependence on time scale. Given the system size, the reviewers judge that it should be easy enough to double the time scale and collect many replicates to ensure that the results are robust. We note that given the issues related to presentation (see below), it is possible that the reviewers have misinterpreted the results and repeats were indeed carried out. For example, there were simulations of System I with different force constants, but those don't seem to be discussed anywhere in the text.

We are sorry for the confusion brought by the original supplementary tables, which contained multiple trial and validation simulations that were not discussed in the main text. We have revised these tables (Supplement File 1a – c), and only kept the production simulations.

In the original manuscript, the simulation time scale was determined by the pulling speed in the MD simulations, and longer simulation time would lead to the disruption of the global protein structure. Following the reviewers’ suggestions, we doubled the simulation time scale by slowing the pulling speed, and performed nine additional 500-ns simulations for the system I (three replicates for each condition, please refer to the revised Supplement File 1a). The simulation results were indeed very robust and consistent with our original observations (Figure 1—figure supplement 2 ).

Also, for system II, which contained more than 1,100,000 atoms, we performed extensive additional simulations with various pulling/pushing forces (2.2μs in total) to obtain more convincing results regarding the force constant of the ankyrin repeats region without disrupting the global structure (please refer to the revised Supplement File 1b and Figure 1—figure supplement 3).

The new simulations for the revision are highlighted in Supplement File 1a – b. We have also revised the corresponding method section and discussed these new simulations in the main text.

Related to the interpretation of the MD simulations, the reviewers had the following comments:The conformational changes observed in Figure 2b,c (and Supplemental Video S1C) are very small (especially panel c). Does this really open the channel? More analysis (with ions moving through the constriction zone for instance or water wetting/dewetting changes) must be done to support this claim. How much water is seen moving back and forth? What about free ion movement into and out of the cavity. Is unbiased ion entry to the cavity observed? Moreover, what about the local pore structure in the pushing experiment shown in Supp Video S1C really starts to open, is it side chain movement or all backbone? What are the backbone radius changes?

Although the conformational changes were not large enough to fully open the channel, the observation of pore dilation was consistent in all of the pushing simulations (Figure 1D and new Figure 1 —figure supplement 2C ). Following the reviewer’s instructive suggestions, we have:

1. Analyzed the water distribution around the narrowest constriction zone, and observed an evident increase of the number of water molecules in the pushing simulations (new Figure 1 —figure supplement 5 ), supporting the notion that a pushing force tends to open the gate and thereby make it easier for water molecules to move back and forth around the gate of the channel.

2. Analyzed ion movement around the gate, and observed two Na^+^ spontaneously passing through the partially opened gate in the pushing simulations without any transmembrane potential (new new Figure 1—figure supplement 6).

3. Analyzed the backbone radius changes, which confirmed that the backbone pore was also dilated under a pushing force (new Figure 1—figure supplement 3 – 4). Thus, the observed pore dilation was not merely a side-chain movement but a global conformational change.

Figure 2d,e – is this based on gazing at the structure itself, or did were these residues identified from changes in the simulations? The authors mention that the values come from the simulations, but one wonders if they could have been guessed from the structure themselves.Similarly, are Figure 3e residues identified from simulation or sequence gazing at the structure? Additionally, the authors state 'we found two stable hydrogen bonds between the ARs and LHs in our MD trajectories, between W1115 and D1142, and R1127 and E1163, respectively'. Cheng and Jan did not find potential hydrogen bonds directly from the atomic model of NOMPC at these two sites structure. The two pairs are either too far to establish a hydrogen bond (D1142-W1115: 5.4 Å) or on the borderline (E1163-R1127: 3.9 Å). The authors should comment on this point – perhaps this points to an important role in the simulations, but why do the simulations show h-bonding while the structure does not?

The H-bonds shown in Figure 2D-2E and Figure 3E were identified from our MD trajectories (Supplement File 1d), most of which were also present in the cryo-EM structure. The agreement between the MD simulations and cryoEM structure was reasonable and can be viewed as cross-validation between the structure and the simulations. The exception was Q1253-S1577, which formed a hydrogen bond for ~50% of our simulation time but not in the cryo-EM structure (Supplement File 1d). It was noticeable that this H-bond was more stable when the AR region was being pushed, while less stable when the AR region was being pulled or free (Supplement File 1b ). Therefore, the formation of this pair of hydrogen bond was stabilized by the pushing force, partially explaining why it was observed in our MD simulation but not in the “force-free” cryo-EM structure.

In the meantime, it was also likely that the MD simulations optimized the local structures. As the reviewer pointed out, we also observed another two stable pairs of H-bonds in our MD simulations that were not present in the cryo-EM structure (D1142-W1115 and E1163-R1127, Figure 3E and Supplement File 1d ), and we think this was due to the fact that MD simulations can dynamically explore the free energy surface around the cryo-EM structure, and thus further optimize the local conformation of the structural model. Considering the resolution of the cryo-EM structure (3.55 Å) was not high enough to see the exact side-chain locations, it was possible that our MD simulations further optimized the model to get a better sampling of the side-chain conformations. As can be seen in the newFigure 3—figure supplement 7. , the distances between the side chains of the above two pairs of hydrogen bond-forming residues obtained from the MD simulations trajectories were closer than those in the cryo-EM structure, while the distances between the α carbon atoms were nearly identical, indicating that the differences observed in the MD simulations and cryo-EM structure were due to the side-chain movements in the simulations. As the MD simulations took into account all the atomistic interactions, including water molecules, ions, lipids, and hydrogen atoms of the protein, we believe the MD simulation result may represent a more optimized model. We have discussed this in the relevant result section.

Looking at the Video S5: in all three simulations illustrated, the AR3-AR4 packing is disrupted. This is not something expected, yet it is not addressed anywhere in the manuscript and should be addressed. This will affect how the forces are transmitted through the protein.

This disruption of the ARs packing was probably caused by the strong and fast pulling forces applied to AR1. To validate, we performed many additional simulations with smaller forces applied to AR1 (Supplement File 1b). In the weaker-force simulations, the AR region was not disrupted, and we obtained a similar force constant of around 3 pN/nm, although with larger error bars (new Figure 3—figure supplement 3 ). We have added a new Video 6 and discussed it in the revised text. That being said, it was also possible that the disruption was due to the absence of the microtubule, which may assist to stabilize the terminal ARs.

The simulations of System II seem to suggest that pushing and pulling on AR1 while restraining the LH domain lead to the LH domain experiencing rotational forces (e.g. Figures 3b and S7). It seems like these are the forces that should be applied to System I, rather than pushing or pulling perpendicularly to the membrane plane.

We projected the forces on the Z-axis and XY plane, and the results showed that the pushing force is still the dominant part of the mechanical stimuli appreciated by the LH domain (new Figure 3—figure supplement 1 ). In the meantime, our results showed that a perpendicular pushing force was enough to drive the channel to open (and the TRP domain to rotate), so we mainly focused on these forces in this work.

The reviewer did raise a very interesting question, and we agree that the rotational forces on the XY plane may assist to drive the TRP domain to move clockwise. This kind of simulation is much more complicated to implement, and we may need to develop a new simulation method to systematically investigate this in future work.

The authors attempt to provide some experimental evidence that NompC is only pushed open by showing NompC failed to respond to negative pressure using cell-attached patch-clamp. However, previously the Jan lab performed the same experiment (Figure5F, Zhang et al., Cell 2015), and their data showed NompC in fact could respond to negative pressure. The authors need to explain the inconsistency between their results and what the Jan lab has shown.

Our results of cell-attached (and inside-out) recordings were different from Zhang et al. (Ref 15, Zhang et al., Cell 2015), while our outside-out recordings showed consistent results with Zhang et al. (Ref 15, Zhang et al., Cell 2015), which confirmed that NompC can be activated by pressure (mechanical forces). We noticed that the resistance of the electrodes used by Zhang et al. was 5-7 MΩ, while ours was 12-20 MΩ. The resistance of our electrode is 2-3 times larger compared to the previous work, which might explain the different observations. In such a case, our electrodes were much thinner, meaning the patched area sucked into the electrodes in our recordings was much smaller than in the previous work. Such a big difference in the patched area may cause different physical changes of the membranes when the pressure is applied. For instance, if the patched area is large, the patched membrane may tend to bend rather than move upwards or downwards along the electrode. On the other hand, if the patched area is small, the patched area can be harder to bend, and therefore it may move upwards and downwards fiercely instead, which may cause direct compressing or stretching of the ankyrin repeat domain. Therefore, we believe our recordings with the smaller patched area may better minimize the effect of membrane curvature, and therefore better represent the effects of stretching or compressing of the ankyrin repeat domains.

The authors state on p. 5, l. 10-102 that "Previous studies have shown that the AR regions are necessary for the mechano-gating of NompC, and the force from lipids cannot open the channel in the absence of the AR region (15)." However, this statement is incorrect: on p. 1401 of ref. 15, the authors of the Zhang et al., 2015 write: "We wish to emphasize that while our results strongly support the notion that ARs function as a tether for mechanogating of NOMPC, our results do not exclude the potential role of interactions between NOMPC protein and the membrane lipids nearby." And indeed, in the 2017 Nature paper by Jin et al. there is evidence that His1423 in the S4-S5 linker of NompC coordinates the lipid polar head group and is required for the channel mechanosensitivity because "the His1423Ala mutant is no longer responsive to mechanical stimulation" (p. 2). And further on p. 4 of this paper: "While it is unlikely that membrane deformation alone is the driving force in NOMPC gating, as in the case of other mechanosensitive channels such as Piezo and TRAAK channels, the proximity of a lipid molecule interacting with the functionally important His1423 at the S4-S5 linker suggests that lipid-protein interactions may have an important role in channel activity." The question is how to reconcile the views of Zhang et al., Jin et al. and the authors of this study, who seem to exclude the possibility of the bilayer involvement in NompC mechanogating? To answer this question, the reviewers have the following suggestion, which can be tested experimentally (and possibly using MD as well) by the authors of this study. Given the unidirectionality of the force causing the channel to open, it is likely that pushing the ankyrin bundle spring would locally curve the membrane bilayer. The local membrane curvature can activate a mechanosensitive channel by changing the asymmetry of the transbilayer pressure profile favouring the channel opening (Cox et al., Cell Reports 29(1): 1-12, 2019). In contrast, the force pulling on the spring would change the local membrane curvature in the opposite direction favouring the closed conformation of the channel. Given that the lipid bilayer is much more compliant compared to a membrane protein the displacement of the NompC channel with respect to the lipid bilayer by pushing the ankyrin spring could trigger the channel open conformation (Kung, Nature 436: 647-654, 2005). As shown by FE modelling (Bavi et al., Membranes 6: 14, 2016) a local curvature corresponding to a radius of less than 50 nm results in membrane tension of several mN/m, which is sufficient to open a mechanosensitive channel. The unidirectionality of the activation force reflecting the pushing vs. pulling the spring in the NompC case can be tested experimentally by inserting an amphipathic molecule, such as OAG (1-oleoyl-2-acetyl-sn-glycerol), a commercially available diacylglycerol (DAG) analogue, which like other amphipathic molecules can bend the cell membrane and was shown to activate the TRPC6 channel, which could not be activated by membrane stretch (Nikolaev et al. (2019) J Cell Sci 132: jcs238360). By adding OAG into the extracellular monolayer of the membrane bilayer it may be possible to open NompC (corresponding to pushing force), whereas adding it to the intracellular monolayer (corresponding to pulling force) would not open the channel. Thus the request from the reviewers is to perform patch clamp experiments using inside-out patch configuration as described in their manuscript. By adding OAG to the extracellular monolayer of the membrane patch accessible from the bath they should be able to open the NompC channel, whereas adding OAG from the pipette they would not be able to do it. Provided the authors could demonstrate this effect of OAG on NompC gating, these experiments together with the results of their study described in this manuscript would reconcile the tethered mechanism with the force-from-lipids paradigm in the case of NompC and indicate that "sharing the force" between the membrane bilayer and cytoskeleton is the likely mechanism of NompC activation by mechanical force.

We thank the reviewers for pointing out this important question. We focused on the mechanical properties of NompC itself and omitted the possible effect of membrane curvature and lateral pressure changes on the gating of NompC, which was indeed flawed. Following the reviewers’ suggestion, we conducted additional patch-clamp recordings with OAG added. Our results showed that the activation was indeed unidirectional: by adding OAG to the intracellular monolayer, the NompC was more likely to be activated; while adding OAG to the extracellular monolayer showed the opposite effect (new Figure 5 E -F ). This is also consistent with the previous study on TRPC6 (Ref 41, Nikolaev et al., J Cell Sci 2019). Given both NompC and TRPC6 belonging to the TRP family, this may imply a common mechanism of the family.

We think this is also consistent with the push-to-open mechanism. Given a relaxed and planar bilayer containing NompC, pushing ARs from the intracellular side would lead to larger lateral stress in the intracellular monolayer than the extracellular monolayer. Adding OAG to the intracellular side would make the intracellular monolayer more crowded, and thus a larger lateral pressure than the extracellular monolayer as well. Therefore, adding OAG to the intracellular monolayer would generate a similar effect on the transmembrane stress profile to applying a pushing force on the AR domain toward the extracellular side. Indeed, it is possible that the pushing force would drive the pore to open and at the same time generate an asymmetry of the transbilayer stress profile that can facilitate gating. It is not clear which is more important, but it is highly likely that both effects work together to open the channel.

We have deleted the statement “the force from lipids cannot open the channel in the absence of the AR region” and edited the text accordingly. Also, we added the new experimental data as well as new text and citations in the Discussion section. We thank the reviewers very much for directing us to give a more comprehensive discussion.

Relatedly, P. 10, l. 205-206: "These results indicate that the interface between the AR and LH regions is also crucial for the force transduction, further confirming the tethered spring model for NompC." – Further to this statement the authors omitted to mention that His1423 residue within the S4-S5 linker was reported to be functionally important for NompC mechanosensitivity.On P. 6, l. 125-128, the authors write: "Thus, consistent with previous structural studies of TRPV1.… it is the pushing force (compression of the intracellular domain) that leads to such a collective gating motion." – This is correct. However, it does not completely explain the mechanism of opening the channel. As stated above, the force opening the channel can also result from a change in the local membrane curvature causing the channel to open given that the force pushing on the ankyrin bundle spring could generate positive curvature and consequently change asymmetry of the transbilayer pressure profile as indicated by the FE modelling study (Bavi et al., Membranes 6: 14, 2016).

Following the reviewer’s suggestion, we further analyzed the interaction between the H1423 and lipid molecules in our MD trajectories. Indeed, it was interesting to see that H1423 can stably interact with a lipid (POPC) molecule in the simulation trajectories, and the lipid molecule was not located at the interaction site in the initial setup but diffused to the right place to act as a bridge to link H1423 and S1268 during the simulations (new Figure 5 A – D ). Therefore, it seems that lipid molecules stabilized the local structure by interacting with H1423, which may be necessary for gating. Also, as discussed above, membrane curvature-induced asymmetrical stress profile may play a role to facilitate gating, and H1423 is probably a key residue for conveying forces from lipids to the NompC channel. We have added the relevant data and rephrased the gating mechanism in the Discussion section to make it more comprehensive.

Additionally, the reviewers judged that many of the conclusions are over-interpreted:Many of the mutations resulted in non-responsive channels. These non-responsive channels could be non-responsive for many reasons, not only gating defects (e.g. folding defects). The authors need to show that the membrane localization and quantity of NompC at the membrane is not affected by the mutations at the least.

Thanks for the suggestion. We performed non-permeablized staining with an antibody targeting the pore-helix (α-NompC-EC) (same as in Ref 15, Zhang et al., Cell 2015) to quantify membranous NompC.

Our staining results showed that for the mutations that led to significant loss-of-function, there were no significant quantity decreases on membranous NompC, except for S1577A and W1572A (Figure 2—figure supplement 5. ). Notably, these two mutations led to a moderate degree of quantity decreases of membranous NompC, but complete loss-of-function in the outside-out recordings, indicating that the two mutations altered the mechnosensitivity of NompC in addition to the quantity of NompC on membranes. Therefore, the results suggested that the point mutations discussed in our work did influence the intrinsic mechanoactivity of NompC.

The other special case is S1421A. As discussed in the text for the case of S1421A, the hydrogen bonding with S1421 is on its backbone rather than side chain, so we expected that the activity of S1421A would not be significantly changed if it can still get into the membrane. Therefore, the observation that the abundance of the membranous S1421A (Figure 2—figure supplement 5. ) and the activity of S1421A (Figure 2F) both decreased was reasonable.

We have added the data in the SI and briefly discussed this in the main text.

The ion conductance in the simulation being lower than experimentally measured is interpreted as the channel conformation being only "partially open". But many other reasons could explain the observed conductance. Are there properties of these currents that you can extract that are not dependent on current or, more importantly, can single channel recordings be performed to confirm the impact on channel function?

We think the difference of the conductances measured from our simulations and experiments is due to the limitation of the simulation time scale, which is a common problem in the MD simulations: the full gating processes of ion channels usually occur on a time scale of millisecondwhile current MD simulations fall on the timescale of hundreds to thousands of nanoseconds, which is not long enough to obtain a fully opened structure, and therefore the ion conductance in the simulations should be lower than that in the fully open state.

Nevertheless, the trend of pore opening caused by the pushing force was very clear and well reproducible in the MD simulations, so we believe the push-to-open gating mechanism is solid; and we conducted further qualitative validations that were not dependent on the specific current values:

1. We conducted cell-attached patch-clamp experiments: from our MD simulation results, we suspected that a positive pressure would activate the NompC channel but not a negative one, and that was what we observed in the cell-attached patch-clamp experiments; adding Gd^3+^ to the bath killed the currents (Figure 1—figure supplement 10), which further confirmed the currents were due to the ion permeation through NompC.

2. By analyzing the interaction networks in MD trajectories, we proposed several potentially important residue pairs involved in the hydrogen bonding that should be important for stabilizing the structure and mechanosensitivity of NompC, and the mutagenesis and electrophysiology experiments validated that these mutations did suppress or abolish the mechanosensitivity of the NompC channel without significantly changing their localization and abundance on membranes ( Figure 2 —figure supplement 5 and Figure 3 – —figure supplement 5).

The single-channel recordings and conductance measurements were performed thoroughly in previous studies by one of our corresponding authors and his colleagues (Ref 9, Yan et al., Nature 2013; Ref 15, Zhang et al., Cell 2015), and they obtained a single-channel conductance of ~150 pS for NompC (~10 pA under 60 mV). This was also observed in our electrophysiology experiments (Figure 5 E- F ).

Taken together, we believe that the difference of conductance values observed in the MD simulations and electrophysiology was due to the common problem of insufficient sampling of MD simulations, but our qualitative results and validations were robust to support the push-to-open gating model.

Hypotheses about compensatory interactions in the hydrogen-bonding networks, etc, are presented as conclusions.

We thank the reviewer for pointing this out. We should have included the analysis to show that the compensatory H-bonding was observed in our simulations. In the revision, we conducted quantitative analysis on these hydrogen bond formations in the trajectories. The results are in the Figure 2—figure supplement 6 and Figure 3—figure supplement 6 and clearly show new H-bonds forming in the mutants during the simulations.

P. 9, L. 191-196: "Considering the fact that the length of the relaxed AR region is about 15 nm, we estimated that the force was transferred through the AR region at a speed of 1.8 ± 0.2 nm/ps.… the force is propagated via the membrane at a speed of 1.4 ± 0.5 nm/ps (37). Therefore, it appears that the force transfer speed in the tethered NompC channel is faster than that in the membranes." This statement requires statistical testing to compare quantitatively the two speeds of force propagation. As presented here, the force propagation through AR region does not appear to be significantly faster from the force propagation via the membrane bilayer. In contrary, they seem quite comparable.

We have revised the text: “Therefore, it appears that the force transfer speed in the tethered NompC channel is comparable to, or slightly faster, than that in the membranes.”

P. 11, l. 232-234: "Nonetheless, a clear and detailed opening process of the channel in the presence of pushing forces was revealed by our MD simulations and validated by electrophysiological experiments." – MD simulations have not been validated experimentally because in outside-out patches CSK connections to the membrane bilayer/channels are largely disrupted, which has been shown to happen upon gigaohm seal formation (see for example Suchyna et al., Biophys J. 97(3): 738-747, 2009). Consequently, opening NompC by pushing the ankyrin spring is not the likely mechanism of the channel opening in membrane patches as suggested by the model shown in Figure 4A of the manuscript.

Here we mainly meant that the cell-attached patch-clamp experiments supported our MD simulation results. We do agree that there are more uncertainties in the outside-out patch-clamp experiments, but the agreement between the MD simulation results, the cell-attached patch-clamp, and the outside-out patch-clamp was reassuring.

We have changed our wording in the text: ‘Nonetheless, a clear and detailed opening trend of the channel in the presence of pushing forces was seen in the MD simulations and this was supported by electrophysiological experiments. The combination of the two methods provides a plausible “push-to-open” gating model for tethered mechanosensitive ion channels.’

The reviewers identified presentation issues that obscured the message of the paper:The methods and Table S1 indicate that three separate simulations with three different force constants were performed. But the authors do not indicate which of these simulations they used in each the figures or analyses. This lack of clarity as to which simulation is used in the analyses is pervasive throughout the manuscript, making a lot of the data difficult to interpret and evaluate. In general, all figure legends or specific mentions of simulations in the main or supplemental text should include a reference to the trajectory labels provided in Tables S1-S3.

Sorry for the confusion and thanks for the suggestions. We have done the following to solve the presentation issues:

1. We revised Supplement File 1a – c to include the production simulations only and labeled all the trajectories.

2. We have included references to the above trajectory labels for all the figure legends. Some of the specific mentions of simulations in the main or supplemental text have also been referenced to the labels when necessary. The other mentions in the text refer to the figures that already have the references, so these are not labeled again.

The concatenation for PCA analysis is fine, but the data should be "unconcatenated" clearly in the figures. This is done in Figure 2A (although using saturation gradients to indicate progression through each of the three simulations would be useful). I.e. in Figure S3, for example, the three different simulations should clearly be demarcated.

We have updated the Figures (Figure 2A and Figure 2 – supplement figure 1 in the revision) to demarcate the trajectories.

The methods section reads more like a mixture of results and methods. The manuscript should be reorganized to separate the two. The methods are also inadequately detailed: Residue numbers should be used to indicate the residues included in various simulations, residues restrained in simulations, residue ranges used as AR1, AR29, LH, TM, etc. The published structure, 5VKQ, is missing residues 1603-1669 between the TRP box and the C-terminal helix that completes the LH domain, how was this handled in building the simulated systems?The methods should provide enough detail that the exact full sequence of the expression vectors (including the codons used for the mutations) can be reconstructed.

Many thanks for these suggestions. we have rewritten the method section by removing the redundant text and moving the relevant results to the main text, and added a new Supplement File 1e and a new Figure 1 – supplement figure 1 to present the residue ranges used in our MD simulations.

The missing residues 1603-1669 were ignored in our MD simulations, as modelling such a large unstable structure would introduce too many uncertainties. The C-terminal helix was removed from the simulation system, as it is located in the large gap between adjacent AR chains, not between the AR and LH domains, and therefore it is expected to play a minor role in the force convey. This was not a perfect solution, but we think the artifacts would be minor.